# A Geometry-Based View of Mahalanobis OOD Detection

**Denis Janiak** [1]  **Jakub Binkowski** [1]  **Tomasz Kajdanowicz** [1]

## Abstract

Out-of-distribution (OOD) detection is critical for reliable deployment of vision models, and Mahalanobis-based detectors remain strong baselines. However, their performance varies widely across modern pretrained representations, making it unclear which feature-space properties determine success or failure. Across diverse foundation-model backbones and Mahalanobis variants, we show that performance is highly representation-dependent and can shift substantially with pretraining data and fine-tuning. We link this variability to in-distribution geometry and identify a two-term ID summary that consistently tracks Mahalanobis OOD behavior across detectors: within-class spectral structure and local intrinsic dimensionality. Finally, we introduce radially scaled $\ell_2$ normalization, $\phi_\beta(z) = z/\|z\|^\beta$, a direction-preserving transformation that changes feature radii and exposes a different ID geometry to the same quadratic detector. Selecting $\beta$ from ID-only geometry signals generally outperforms fixed normalization baselines.

## 1. Introduction

Mahalanobis scores are among the simplest post-hoc detectors for out-of-distribution (OOD) detection, yet they remain surprisingly competitive on modern vision backbones (Lee et al., 2018; Mueller & Hein, 2025; 2024). At the same time, their behavior is highly representation-dependent: the same quadratic detector can perform well on one pretrained model and fail on another, and performance can shift sharply with changes in pretraining data or fine-tuning regime. This sensitivity makes Mahalanobis-based OOD detection difficult to deploy reliably and raises a basic question: *which properties of an in-distribution feature space determine when a Mahalanobis detector succeeds or fails?*

We study Mahalanobis OOD detection through the lens of representation geometry. Across self supervised and foundation model representations, we show that geometric structure accounts for much of the observed cross model variation. In particular, a compact ID-only summary combining local intrinsic dimensionality and within-class spectral decay strongly predicts Mahalanobis OOD performance across variants. This connects detector reliability to measurable properties of the in-distribution feature space.

Motivated by this geometric view, we introduce a simple post-hoc control mechanism that changes the geometry presented to the same quadratic detector. We use radially scaled $\ell_2$ normalization, $\phi_\beta(z) = z/\|z\|^\beta$, which preserves feature directions while contracting or expanding radii. Unlike prior work that modifies the scoring rule (Ren et al., 2021) or fixes normalization to the unit sphere (Mueller & Hein, 2025), varying $\beta$ provides a continuous way to reshape radial geometry without altering the detector form. Figure 1 illustrates how $\beta$ tightens or spreads decision regions by changing feature radii. Empirically, adjusting $\beta$ induces structured, model-specific changes in both geometry and OOD performance. Leveraging the same ID-only geometry signals, we propose a practical procedure to select $\beta$ without access to OOD samples, often improving over fixed baselines such as $\beta = 0$ (standard features) and $\beta = 1$ (unit-sphere normalization).

Our contributions are: **(i)** a broad benchmark of Mahalanobis-style OOD detectors across diverse SSL/foundation models, including a per-dimension analysis of detector behavior; **(ii)** an empirical link between Mahalanobis OOD performance and ID geometry, with an ID-only summary that consistently predicts performance across detector variants; and **(iii)** a geometric control mechanism via $\beta$-scaled radial normalization, together with an ID-only $\beta$ selection rule that approaches oracle tuning without requiring OOD access.

## 2. Related Work

OOD detection is essential for ensuring the reliability of machine learning systems in real-world deployment (Fort et al., 2021). Its goal is to identify whether inputs stem from the training distribution, thus preventing overconfident predictions on unexpected data (Yang et al., 2024).

[1]Department of Artificial Intelligence, Wrocław University of Science and Technology, Wrocław, Poland. Correspondence to: Denis Janiak <denis.janiak@pwr.edu.pl>.

*Proceedings of the 43ʳᵈ International Conference on Machine Learning*, Seoul, South Korea. PMLR 306, 2026. Copyright 2026 by the author(s).

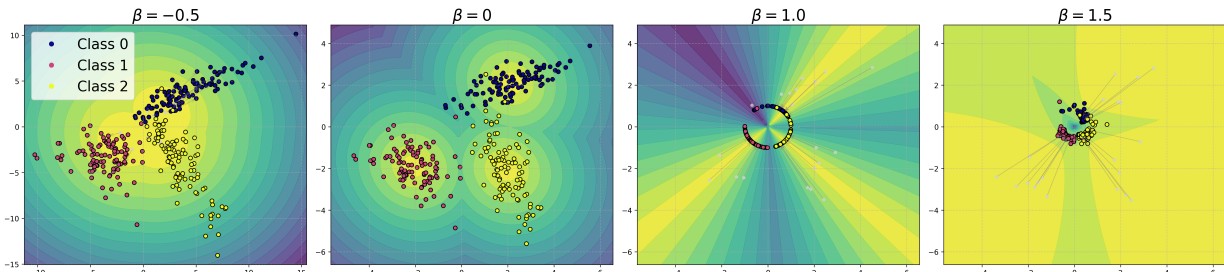

*Figure 1.* **Effect of radially scaled $\ell_2$ normalization on feature geometry and Mahalanobis boundaries.** Visualization in 2D of the induced geometry and the resulting Mahalanobis decision regions. Gray arrows indicate the radial mapping from the original to the transformed space. Larger $\beta$ contracts feature norms and tightens clusters, producing more localized decision regions; smaller $\beta$ expands norms and broadens them. Choosing $\beta$ appropriately can reduce ID–OOD overlap and improve OOD detection.

Post-hoc, training-free methods are particularly effective, as they combine efficiency with robustness without altering the model (Xu et al., 2023). Among OOD detection methods, Mahalanobis distance has become a cornerstone (Lee et al., 2018), with several refinements improving its robustness and performance. The standard Mahalanobis distance (MD) uses class-conditional covariance estimates to measure the distance of a sample from each class mean. In contrast, the Relative Mahalanobis distance (RMD) (Ren et al., 2021) compares each class-specific distance to a single global Gaussian fitted to all in-distribution (ID) data, effectively normalizing class distances against a global reference. Mahalanobis++ (Mueller & Hein, 2025) further improves performance by L2-normalizing features, making them adhere more closely to the Gaussian assumptions underlying the Mahalanobis distance. However, our study reveals broader insight into the influence of normalization when computing Mahalanobis distance, particularly in the context of vision models.

Vision OOD detection has shifted toward leveraging large-scale pretraining and contrastive objectives, where vision transformers (Dosovitskiy et al., 2021) and CLIP (Radford et al., 2021) show strong near-OOD performance and benefit markedly from few-shot outlier exposure and even label-only supervision for outlier classes (Fort et al., 2021). However, full fine-tuning can distort pretrained representations and harm OOD generalization relative to linear probing, with similar cautions for vision–language models; recent work also explores training-time scaling and post-hoc enhancements, and revisits detector design in vision foundation models (Fort et al., 2021; Ming & Li, 2024; Xu et al., 2023; Zhao et al., 2024b). Evaluation rigor has improved through ImageNet-scale suites like NINCO that mitigate in-distribution leakage. Meanwhile, theory and diagnostics connect feature separability to OOD error and delineate when OOD detection is learnable (Bitterwolf et al., 2023; Xie et al., 2023).

Representation geometry and normalization have attracted increased attention for their role in OOD generalization.

An analysis of contrastive learning and normalization approaches (Le-Gia & Ahn, 2023; Tan et al., 2025) shows that geometric priors, such as hyperspherical projection or $\ell_2$ normalization, can yield more robust representation spaces. Studies like (Zhao et al., 2024a) and (Xie et al., 2023) link improved feature separability and lower intrinsic dimensionality to higher OOD detection performance.

## 3. Background: Mahalanobis OOD Detection

Let $z = f(x) \in \mathbb{R}^d$ denote the feature representation of an input $x$. Given ID training features from $K$ classes, Mahalanobis-based detectors model each class by a Gaussian $\mathcal{N}(\mu_k, \Sigma)$ with a *tied* covariance $\Sigma$ (the LDA assumption), and optionally a marginal Gaussian $\mathcal{N}(\mu_0, \Sigma_0)$ fitted to all ID features.

**Mahalanobis variants.** The class-conditional Mahalanobis distance (MD) and its confidence score are

$$\begin{aligned} \mathrm{MD}_k(z) &= (z - \mu_k)^\top \Sigma^{-1}(z - \mu_k), \\ \mathcal{C}_{\mathrm{MD}}(x) &= -\min_k \mathrm{MD}_k\big(f(x)\big). \end{aligned} \tag{1}$$

Marginal Mahalanobis (MMD) uses the class-agnostic quadratic form $\mathrm{MD}_0(z) = (z - \mu_0)^\top \Sigma_0^{-1}(z - \mu_0)$ as its score, and Relative Mahalanobis (RMD) (Ren et al., 2021) subtracts this marginal reference:

$$\begin{aligned} \mathrm{RMD}_k(z) &= \mathrm{MD}_k(z) - \mathrm{MD}_0(z), \\ \mathcal{C}_{\mathrm{RMD}}(x) &= -\min_k \mathrm{RMD}_k(z). \end{aligned} \tag{2}$$

**Eigenbasis view.** Let $\Sigma = U\Lambda U^\top$ have eigenvalues $\lambda_1 \geq \cdots \geq \lambda_d > 0$ and eigenvectors $\{u_i\}$. Define the per-direction energy $\tilde{a}_i(z) \triangleq (u_i^\top(z - \mu_k))^2$. Then

$$\mathrm{MD}_k(z) = \sum_{i=1}^d \lambda_i^{-1} \tilde{a}_i(z). \tag{3}$$

This form highlights how directions with smaller variance (smaller $\lambda_i$) receive larger inverse weighting and motivates

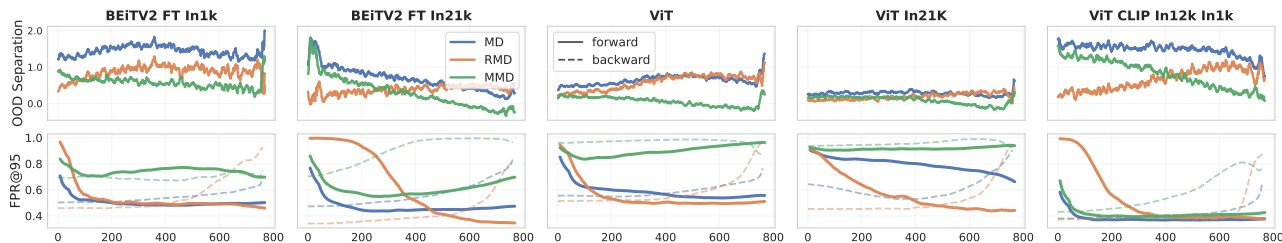

*Figure 2.* **Dimension-wise OOD behavior for Mahalanobis variants.** The top row shows dimension-wise OOD separation $S_i$, and the bottom row reports FPR under progressive dimension ablation. Strong embedding-space separation does not necessarily translate to better OOD detection.

our later analysis of how representation geometry influences Mahalanobis-style OOD behavior.

# 4. Comparative Study of Foundation Models

## 4.1. Cross-model OOD Detection Performance

We begin by characterizing how representation learning choices shape Mahalanobis-style OOD detection across modern vision backbones. In particular, we ask how performance depends on architecture, pretraining data, and fine-tuning regime, i.e., factors whose effects are not systematically documented. This motivates a broad, model-agnostic comparison: *Which modern self-supervised or pretrained vision models produce representations that naturally lend themselves to Mahalanobis-style OOD detection?*

**Evaluation protocol.** Following OpenOOD (Yang et al., 2022), ImageNet-1K serves as the in-distribution (ID) dataset (train features for fitting; validation for ID testing). We report FPR@95 for distinguishing ImageNet validation from each of five OOD benchmarks: NINCO (Bitterwolf et al., 2023), iNaturalist (Van Horn et al., 2018), SSB-Hard (Bitterwolf et al., 2023), OpenImages-O (Krasin et al., 2017), and Textures (Cimpoi et al., 2014). Unless stated otherwise, we fit class means $\{\mu_k\}$ and a tied covariance $\Sigma$ on ImageNet-1K training features and evaluate OOD scores on the ImageNet validation set versus each OOD dataset. We use publicly available checkpoints from `timm` (Wightman, 2019) and `huggingface-transformers` (Wolf et al., 2020), covering several transformer families and training regimes, including BEiT-v2, CLIP, EVA-02, and ViT variants with different pretraining and fine-tuning configurations. The full model list is provided in Appendix L.

**Results (Figure 3).** RMD improves over the standard Mahalanobis distance in most settings, with the largest gains for models that are pretrained but not fine-tuned on ImageNet. Notably, RMD substantially improves OOD detection for EVA02-In21k and ViT-In21k, in some cases matching or exceeding the performance of their ImageNet-fine-tuned counterparts. This weakens the typical association between in-distribution accuracy and FPR, and produces more consistent score distributions across models. At the same time, classification accuracy is not a reliable proxy for OOD performance: large accuracy gaps (often $> 10\%$) do not necessarily yield better detection. We observe only a mild trend along the fine-tuning sequence In1k $\to$ In22k-In1k $\to$ larger In22k-In1k models; full results are reported in Appendix G.

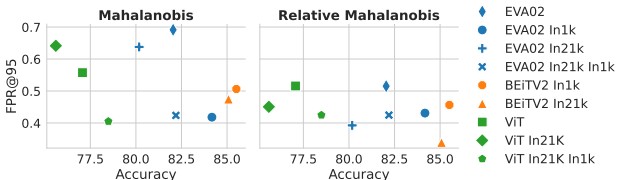

*Figure 3.* **OOD detection performance on NINCO across model families.** RMD consistently improves over standard MD, with the largest gains for models that are pretrained but not fine-tuned on ImageNet.

## 4.2. Mahalanobis Variants and Per-Dimension Analysis

Having established cross-model trends, we next ask: *which parts of the representation space drive OOD discrimination?* Beyond aggregate scores, inspecting individual eigendirections helps explain why some models detect OOD reliably while others do not. We therefore analyze MD, MMD, and RMD at the level of their *per-direction* contributions.

**Per-direction separation.** Using the decomposition in Eq. 3, we define the OOD separation in the eigen-direction $i$ as the difference between its mean contribution to OOD and ID:

$$S_i \triangleq \lambda_i^{-1}(\mathbb{E}_{x \sim \mathcal{D}_{\text{OOD}}}[\tilde{a}_i(x)] - \mathbb{E}_{x \sim \mathcal{D}_{\text{ID}}}[\tilde{a}_i(x)]), \quad (4)$$

where the projection term $\tilde{a}_i(\cdot)$ utilizes the selected class $k \in \arg\min_c \text{MD}_c$ (or $\arg\min_c \text{RMD}_c$ for RMD; see Eq. 1 and 2). $S_i$ measures the difference between average ID and OOD Mahalanobis energy along eigen-direction $i$. Positive $S_i$ indicates that OOD samples contribute more inverse-variance weighted energy than ID samples along $u_i$. We order eigenvalues as $\lambda_1 \geq \cdots \geq \lambda_d$, so larger $i$ corresponds to smaller-variance directions.

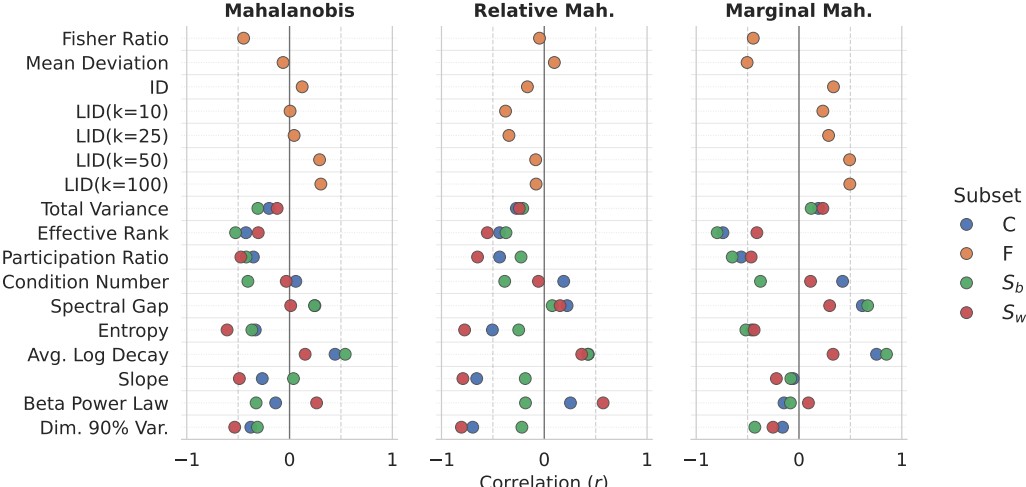

*Figure 4.* **Correlations between representation metrics and OOD performance.** Spearman correlations between representation metrics and OOD performance across Mahalanobis variants. Here, **F** denotes metrics computed directly from feature representations, whereas **C**, $S_b$, and $S_w$ are derived from the eigenvalue spectra of the covariance/scatter matrices (Appendix A). The three Mahalanobis-based detectors leverage distinct geometric cues, yielding different correlation patterns. Similar trends hold for Pearson correlations (Appendix J).

**Ablation protocol.** Figure 2 (bottom) reports an ablation study where we recompute the FPR using only the first $q$ eigen-directions (forward ablation) or only the last $q$ directions (backward ablation). This isolates whether discrimination arises from high-variance structure (small $i$) or from low-variance components that receive strong inverse-variance weighting (large $i$).

**Results.** Figure 2 reveals three recurring behaviors across models. (i) Large per-direction separation does not necessarily yield low FPR: for example, BEiTV2 FT In1k exhibits stronger separation across many directions yet performs on par with (or worse than) BEiTV2 FT In21k. (ii) The effective number of eigen-directions needed for strong detection varies substantially: some models saturate quickly with small $K$, while others require most of the spectrum to approach their best FPR. (iii) Backward ablation shows that low-variance directions can dominate discrimination in some settings; e.g., ViT In21k attains its best FPR primarily from the latter part of the spectrum, suggesting that small-variance components can carry disproportionate OOD signal after inverse-variance weighting. These observations motivate our later stability analysis: performance is governed not only by mean ID–OOD separation but also by how quadratic weighting interacts with the representation spectrum and the allocation of sample energy across eigen-directions.

## 5. Geometry of Representations

In the previous section, we showed that no single OOD method yields consistent performance and behavior across multiple models. In fact, different SSL models and pretraining regimes produce representations with distinct geometric

properties, indicating that OOD performance depends on the intrinsic structure of the representation space. To understand these effects, we analyze the internal geometry of model representations, seeking to answer: *What internal characteristics of a model's feature space predict strong OOD detection?*

### 5.1. Geometry-detector Alignment

To identify which representation properties matter for OOD detection, we correlate detection performance with two complementary families of metrics: (i) **manifold metrics**, such as intrinsic dimensionality (Ma et al., 2018), computed from the ID features $F$, and (ii) **spectral metrics** computed from the eigenspectra of the global covariance $C$ and Fisher scatter matrices $S_w$, $S_b$ (see Appendix A for definitions).

Figure 4 summarizes Spearman correlations across detectors and models. As we can see, the RMD correlates most strongly with within-class geometry ($S_w$), reflecting the importance of compact, well-structured class clusters. MMD correlates primarily with global geometry ($C$ and $S_b$), indicating dependence on the overall manifold shape. Standard MD sits between these extremes, combining sensitivity to both cluster structure and the global eigenspectrum. In Appendix D, we provide a spectral analysis that offers insight into how pretraining and fine-tuning shape these characteristics.

### 5.2. Ideal Geometry: A Compensatory Trade-off

OOD detection depends not only on mean ID-OOD separation but also on how *ID variability is organized.* Two geometric factors repeatedly appear across models: **local**

**degrees of freedom** (how many directions are explored in a neighborhood) and **within-class concentration** (how tightly class clusters concentrate around their means). We quantify each factor with a single scalar. *Local intrinsic dimensionality* measures the effective dimensionality of the feature manifold around a point $z$ via the $k$-nearest-neighbor MLE (Ma et al., 2018):

$$\text{LID}_k(z) = -\left( \frac{1}{k} \sum_{j=1}^{k} \log \frac{r_j(z)}{r_k(z)} \right)^{-1},$$

where $r_j(z)$ is the distance to the $j$-th nearest neighbor; we report the dataset average $m_k = \mathbb{E}_z[\text{LID}_k(z)]$ with $k=50$ throughout (sensitivity in Appendix B). The *within-class spectral slope $s$* is the least-squares slope of the log-eigenvalue spectrum of the within-class scatter matrix $S_w$; its magnitude $|s|$ measures the rate of spectral decay and thus the concentration of within-class variance along a few dominant directions.

Intuitively, $m_k$ captures manifold richness while $|s|$ reflects class compactness: if the local manifold is simple (low $m_k$), reliable detection requires very concentrated clusters (high $|s|$); if the local manifold is richer (high $m_k$), OOD samples can deviate along many directions, relaxing the compactness requirement. This yields a compensatory trade-off between local dimension and within-class concentration. Empirically, this trade-off is captured by the product $m_k \cdot |s|$. Figure 5 shows that $m_k|s|$ strongly predicts Mahalanobis OOD performance across models and variants.

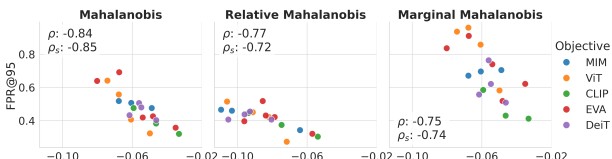

*Figure 5.* **A simple predictor of Mahalanobis OOD performance.** The product $m_k|s|$ (LID × within-class spectral-slope magnitude) correlates with Mahalanobis-based OOD detection across variants.

Taken together, these results suggest that neither local dimensionality nor within-class concentration alone is sufficient to explain Mahalanobis OOD behavior. Instead, strong performance is associated with representations that jointly balance these two properties, for which $m_k|s|$ provides a compact summary. This suggests that if we can move a representation along the $m_k \times |s|$ trade-off curve post-hoc, we may tune Mahalanobis behavior without changing the backbone. In the following section, we show how this summary can be *traced and optimized* by a simple post-hoc deformation of the feature space and later provide a mechanistic explanation for why $m_k|s|$ is predictive across Mahalanobis variants.

## 6. Radial Scaling as a Geometric Control Knob

We introduce a post-hoc, one-parameter family of **direction-preserving radial deformations** that modifies ID geometry without changing the backbone. This family generalizes $\ell_2$ normalization and provides a continuous knob $\beta \mapsto (m_k(\beta), s(\beta))$ using *ID data only*.

### 6.1. Why Radial Normalization?

Mahalanobis-style detectors are quadratic forms that depend on a shared covariance estimate. In practice, feature norms can vary substantially across samples and models, and this radial variability can dominate the covariance fit and inflate score overlap between ID and OOD. Prior work has shown that $\ell_2$ normalizing features (projecting to the unit sphere) can stabilize Mahalanobis OOD detection by reducing norm-driven variation and improving the fit of quadratic scores (Mueller & Hein, 2025). We choose a direction-preserving radial family because it is the minimal intervention that changes norm-driven geometry while preserving angular class structure.

### 6.2. Radially Scaled Mahalanobis Distance

Given a feature vector $z \in \mathbb{R}^d \setminus \{0\}$, we define the radial map

$$\phi_\beta(z) = \frac{z}{\|z\|^\beta}, \tag{5}$$

where $\beta \in \mathbb{R}$ controls radial contraction/expansion while preserving direction. The induced radius is $\|\phi_\beta(z)\| = \|z\|^{1-\beta}$. Thus, for $\beta > 1$, the map contracts norms greater than 1 and expands norms smaller than 1 (pushing toward the unit sphere); $\beta < 1$ has the opposite tendency (pushing away from the unit sphere); and $\beta < 0$ expands the norms (see Appendix E for more details). This family contains key special cases: $\beta = 0$ recovers the original geometry, and $\beta = 1$ projects features onto the unit sphere. In practice, most feature norms are $> 1$. Figure 1 provides a 2D schematic of the induced radial mapping and its effect on the resulting quadratic boundaries.

**Definition.** We denote by **RS-MD** the Mahalanobis-distance detector applied to features after the transformation $\phi_\beta$ in Eq. 5. Thus, $\beta = 0$ recovers standard MD, and $\beta = 1$ corresponds to the $\ell_2$-normalized variant (MD++). We define **RS-RMD** analogously for the relative Mahalanobis variant evaluated on $\phi_\beta(z)$. For each value of $\beta$, the corresponding class means and covariance estimates are refit in the transformed feature space, so $\beta$ changes the geometry seen by an otherwise fixed scoring rule.

*Table 1.* **OOD detection (FPR@95, ↓)** averaged over five OpenOOD datasets for the compact main-paper model subset. *MD* uses features as-is ($\beta = 0$), *MD++* applies $\ell_2$ normalization ($\beta = 1$). *RS-MD* selects a per-model radial exponent $\hat{\beta}$ from ID data by selecting optimal geometric proxy $P(\beta) = m(\beta)|s(\beta)|$ over the search grid. *RMD* denotes the relative Mahalanobis variant, with analogous *RMD++* ($\beta = 1$) and *RS-RMD* (proxy-selected $\hat{\beta}$) settings. For each model (column), *RS-MD* is highlighted in light green when it improves over both *MD* and *MD++*; likewise, *RS-RMD* when it outperforms both *RMD* and *RMD++*. Best (**lowest**) results in each column are bolded.

| Detector | BEiTv2 In1k | ViT In1k | ViT In21k→In1k | ViT-L In21k→In1k | DeiT3 In1k | DeiT3 In21k→In1k | DeiT3-L In22k→In1k | EVA02 In1k | EVA02 In21k→In1k | CLIP In1k | CLIP In12k→In1k | CLIP-L In12k→In1k | Avg |
|---|---|---|---|---|---|---|---|---|---|---|---|---|---|
| MSP | 52.2 | 56.5 | 53.7 | 44.8 | 55.0 | 56.7 | 58.1 | 53.2 | 53.0 | 55.2 | 49.0 | 45.0 | 52.7 |
| MLS | 50.7 | 50.4 | 40.7 | 29.8 | 59.2 | 64.3 | 65.9 | 55.3 | 58.9 | 65.3 | 51.9 | 43.6 | 53.0 |
| KNN | 42.6 | 50.0 | 47.7 | 34.3 | 47.5 | 37.0 | 35.9 | 40.6 | 42.3 | 41.1 | 32.5 | 30.1 | 40.1 |
| VIM | 39.3 | 53.0 | 36.1 | **25.0** | 47.2 | 37.5 | 39.7 | 43.9 | **37.1** | 41.7 | 30.0 | 28.0 | 38.2 |
| RS-MD | **37.2** | 45.5 | 35.8 | 25.3 | 43.2 | **35.5** | **33.4** | **37.2** | 39.5 | **37.6** | **26.4** | **26.7** | **35.3** |
| MD++ | 37.6 | 45.4 | 38.7 | 28.2 | 43.0 | 35.6 | 34.2 | 37.4 | 38.2 | 38.2 | 27.8 | 27.1 | 36.0 |
| MD | 40.2 | 45.7 | **35.7** | 25.3 | 43.3 | 37.6 | 36.6 | 37.6 | 40.8 | 40.2 | 33.5 | 29.7 | 37.2 |
| RS-RMD | **37.1** | 44.8 | 37.5 | 26.9 | 39.4 | 34.5 | 35.5 | 39.6 | 38.6 | 38.2 | 30.7 | 27.0 | 35.8 |
| RMD++ | 37.3 | **44.6** | 37.6 | 26.9 | 39.9 | 35.1 | 35.9 | 39.8 | 39.1 | 38.6 | 30.9 | 27.7 | 36.1 |
| RMD | 39.1 | 44.9 | 37.6 | 26.9 | 40.8 | 36.6 | 37.6 | 40.3 | 40.3 | 40.3 | 32.5 | 29.3 | 37.2 |

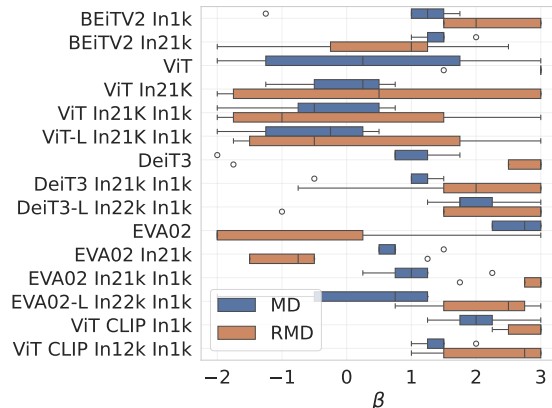

*Figure 6.* **Empirically optimal $\beta$ varies across OOD settings.** Distribution of the empirically optimal $\beta$ for MD and RMD detectors across OOD datasets. The wide spread highlights substantial model- and dataset-specific variation, indicating that $\beta$ typically requires tuning per setting.

## 6.3. Why $\beta$ Needs Tuning?

Applying $\phi_\beta$ systematically alters two aspects of the representation: (i) local neighborhood structure, captured by LID $m_k(\beta)$, and (ii) the within-class scatter spectrum $S_w(\beta)$, captured by the slope $s(\beta)$. Both feed back into the quadratic score through the refit covariance, so the $\beta$ that best aligns features with the tied-Gaussian assumptions of Mahalanobis scoring is model- and dataset-dependent. Moderate positive values often improve this alignment, but in some cases larger or even negative values yield stronger in/out-of-distribution separation, so a fixed choice of $\beta$ is rarely optimal. Figure 6 summarizes the distribution of empirically optimal $\beta$ values (searched over $[-2, 2]$ in 0.25 steps) for MD and RMD across OOD datasets; the wide spread confirms that a one-size-fits-all choice is ineffective.

## 6.4. ID-Only Selection of $\beta$ via a Geometric Proxy

**Setup.** We instantiate the geometry proxy on the same backbones, ID/OOD datasets, and evaluation protocol as

Section 4.1. For each backbone and each $\beta$ on a grid $\mathcal{B} \subset [-2, 2]$ with step 0.25, we estimate two quantities on a held-out portion of the ImageNet-1K training features in the $\phi_\beta$-space: the dataset-average LID $m(\beta)$ with $k=50$ (Section 5.2), and the within-class spectral slope $s(\beta)$ from the eigenspectrum of $S_w(\beta)$. We combine them into a single ID-only geometry summary,

$$P(\beta) \triangleq m(\beta)\,|s(\beta)|, \qquad (6)$$

and evaluate $P(\beta)$ over the grid. The selected $\hat{\beta}$ is then applied unchanged for that model on ImageNet (ID) and on all OOD benchmarks. Figure 7 illustrates this rule on representative NINCO curves, showing both the proxy-selected $\hat{\beta}$ and the oracle $\beta^*$ that minimizes FPR@95.

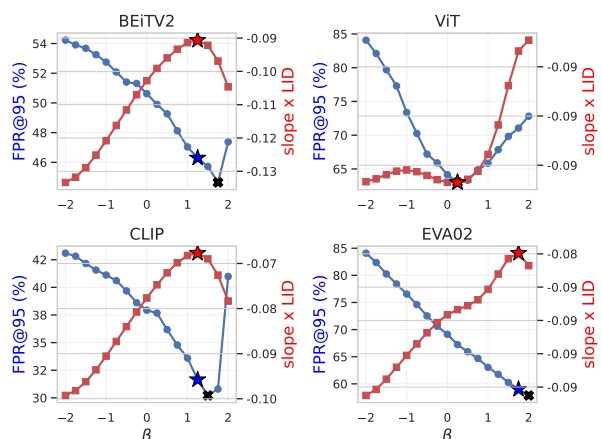

*Figure 7.* **Proxy feature–FPR trade-off across $\beta$ on NINCO (standard MD).** Each panel corresponds to a model configuration. The blue curve reports the FPR@95 as a function of $\beta$, while the red curve (right axis) shows the in-distribution proxy feature $P(\beta) \triangleq m(\beta)|s(\beta)|$. The star indicates the $\beta$ selected by the interior turning-point rule applied to the proxy curve, and the black x denotes the oracle $\beta$ that minimizes FPR@95.

**Selecting $\hat{\beta}$ from the proxy curve.** Across models, $P(\beta) = m_k(\beta)|s(\beta)|$ typically has an interior turning

point (often inverted-U; occasionally U-shaped, e.g., ViTs). Since boundary optima can be artifacts of the finite search range, we select the most pronounced *interior* turning point: the interior grid value farthest from the endpoint baseline $P_{\text{end}} = (P(\beta_{\min}) + P(\beta_{\max}))/2$ (Appendix E.4). This recovers the interior maximum for inverted-U curves and the interior minimum for U-shaped curves; if no clear interior turning point appears on the grid, we use the fallback described in Appendix E.4. In Section 7.2, we connect these regimes to the instability functionals $\mathcal{I}(\beta)$ and $\widehat{\mathcal{I}}(\beta)$. Note that Section 5.2 reports an across-model correlation at $\beta = 0$, which need not determine the within-model optimum along $\beta$.

**OOD performance summary.** Table 1 reports FPR@95 across backbones and OOD datasets. The proxy-selected $\hat{\beta}$ consistently improves OOD detection relative to fixed choices (e.g., $\beta = 0$ for standard MD and $\beta = 1$ for MD++), for both MD and RMD. The top panel of Table 2 summarizes the same comparison by near- and far-OOD split against the fixed $\beta = 1$ baseline: proxy selection improves raw FPR@95 for both MD and RMD in each split, with the strongest absolute gains in the near-OOD regime. Although the proxy does not always match the empirically optimal $\beta$, it captures enough ID geometric structure to yield meaningful gains in detection performance.

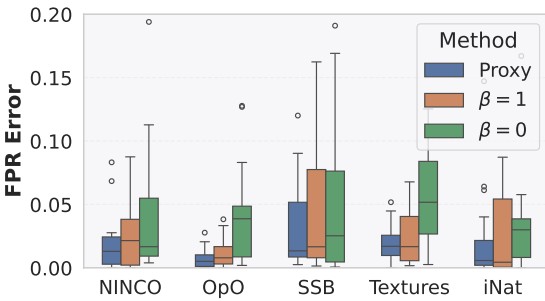

*Figure 8.* **Absolute FPR error relative to the oracle $\beta^*$ (standard MD).** Boxplots show $|\text{FPR}(\hat{\beta}) - \text{FPR}(\beta^*)|$ across OOD benchmarks. The proxy-selected $\hat{\beta}$ consistently achieves lower error than the fixed baselines $\beta=0$ and $\beta=1$ on every dataset.

Figure 8 further quantifies selection quality for standard MD by plotting oracle regret, $|\text{FPR}(\hat{\beta}) - \text{FPR}(\beta^*)|$, across OOD benchmarks and the same set of models as in Table 1. The proxy achieves lower regret than the $\beta = 0$ and $\beta = 1$ baselines across datasets, with strong improvements on NINCO, whose samples were verified to be free of ID contamination. The proxy also reduces worst-case behavior, as reflected by a lower upper tail in the regret distribution. Full OOD results and comparisons against baseline detectors are provided in Appendix G; a more detailed selector-focused regret analysis, including near/far regret, per-dataset breakdowns, $k$-robustness, and alternative geometric selectors, is provided in Appendix H.

*Table 2.* **Near- and far-OOD performance for proxy-based $\beta$ selection.** Raw FPR@95 is reported separately for MD and RMD, comparing the proxy-selected $\hat{\beta}$ against the fixed $\beta = 1$ baseline. *Near-OOD*: NINCO, SSB-Hard. *Far-OOD*: iNaturalist, OpenImages-O, Textures. Lower is better throughout.

| Raw FPR@95 by detector and OOD split | | | | |
|---|---|---|---|---|
| Detector | Selector | Far-OOD | Near-OOD | All |
| MD | $\beta = 1$ | 20.63 | 58.95 | 35.96 |
| MD | **Proxy** | **20.23** | **57.82** | **35.26** |
| RMD | $\beta = 1$ | 21.68 | 57.76 | 36.11 |
| RMD | **Proxy** | **21.36** | **57.49** | **35.81** |

## 7. Unified Stability Theory for Mahalanobis Variants

The previous sections establish two empirical facts: (i) Mahalanobis-style OOD detection varies widely across representations, and (ii) the ID-only geometry summary $m_k(\beta) |s(\beta)|$ tracks performance across models and along $\beta$ trajectories. This section separates the algebraic part of this observation from its empirical interpretation.

Our algebraic starting point is an exact identity for any *single-quadratic* Mahalanobis score: the score separates into a *size* factor, measuring the residual norm, and a *stretch* factor, measuring how whitening redistributes that residual across eigendirections. The subsequent links between these channels, local intrinsic dimensionality, spectral slope, and FPR are empirical: they summarize the patterns and interventions observed in our evaluated model suite rather than distribution-free consequences of the identity.

### 7.1. Score Factorization and Instability Decomposition

Let $\delta_\beta(z) \in \mathbb{R}^d$ denote the detector-specific centered deviation in $\phi_\beta$-space (e.g., class-conditional for MD, global for MMD), and let $\Sigma(\beta) \succ 0$ be the tied ID scatter used by the detector after the same numerical regularization used for covariance inversion in our experiments.

**Proposition 7.1** (Size–stretch identity for a single quadratic score)**.** *Fix $\beta$ and assume that $\Sigma(\beta) \succ 0$ and $\|\delta_\beta(z)\| > 0$ almost surely under the distribution over which the score is evaluated. For*

$$S_\beta(z) = \delta_\beta(z)^\top \Sigma(\beta)^{-1} \delta_\beta(z), \qquad (7)$$

*define the whitened stretch factor*

$$W_\beta(z) \triangleq \frac{\delta_\beta(z)^\top \Sigma(\beta)^{-1} \delta_\beta(z)}{\|\delta_\beta(z)\|^2}. \qquad (8)$$

*Then*

$$S_\beta(z) = \|\delta_\beta(z)\|^2 \, W_\beta(z). \qquad (9)$$

*Consequently, if $\log S_\beta(z)$ has finite variance over ID sam-*

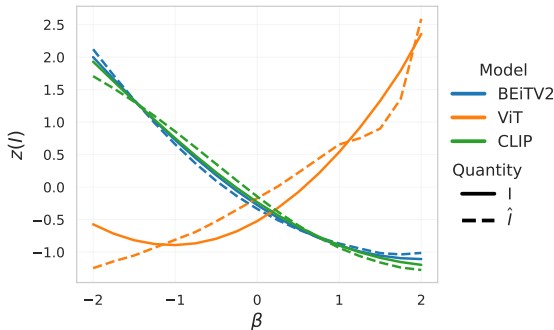

*Figure 9.* **UST proxy tracks instability along $\beta$.** For representative models, we plot $\mathcal{I}(\beta) = \text{Var}(\log S_\beta)$ (solid) and the ID-only proxy $\widehat{\mathcal{I}}(\beta)$ (dashed), both standardized within each model across $\beta$. The proxy tracks the shape of the instability trajectory and predicts its minimizer $\hat{\beta}$ using ID-only quantities. Summary statistics across all models appear in Appendix F.4.

*ples, the instability functional*

$$\mathcal{I}(\beta) \triangleq \text{Var}_{z \sim \mathcal{D}_{\text{ID}}}\big[\log S_\beta(z)\big] \qquad (10)$$

*decomposes as*

$$\mathcal{I}(\beta) = A_\delta(\beta) + A_W(\beta) + 2A_\times(\beta), \qquad (11)$$

*where* $A_\delta = \text{Var}[\log \|\delta_\beta\|^2]$, $A_W = \text{Var}[\log W_\beta]$, *and* $A_\times = \text{Cov}(\log \|\delta_\beta\|^2, \log W_\beta)$.

*Proof.* The factorization follows directly by substituting the definition of $W_\beta(z)$ into Eq. 9. Taking logarithms gives $\log S_\beta(z) = \log \|\delta_\beta(z)\|^2 + \log W_\beta(z)$. Applying $\text{Var}(X + Y) = \text{Var}(X) + \text{Var}(Y) + 2\,\text{Cov}(X, Y)$ yields Eq. 11. $\square$

The identity isolates two ID-side channels. The size channel $A_\delta$ measures dispersion in residual norms, while the stretch channel $A_W$ measures how strongly whitening amplifies different residual directions. Empirically, $A_\times$ is predominantly negative (median log-correlation $\approx -0.528$): large-norm residuals often concentrate along high-variance eigendirections, which carry smaller whitening weights $1/\lambda_i$ and therefore contribute less to $W_\beta$. This sign pattern is an empirical regularity of the evaluated representations, not an assumption of the identity.

The stretch channel is tied to spectral heterogeneity of $\Sigma(\beta)$: steep eigenvalue decay makes the whitening weights $\{1/\lambda_i\}$ heterogeneous, so $W_\beta(z)$ depends strongly on which eigenmodes residuals occupy. The size channel is associated with local feature-space dimension: larger $m_k$ tends to coincide with more diffuse feature support and greater dispersion of residual norms. Appendix F verifies the exact identity numerically and reports isolating interventions for these empirical channel interpretations.

## 7.2. From Instability Channels to a Geometry Proxy

The decomposition in Eq. 11 motivates pairing one ID-only statistic with each empirical channel: $|s(\beta)|$, the magnitude of the log-eigenvalue slope of $\Sigma(\beta)$, for the stretch channel; and $m_k(\beta)$, the $k$NN local intrinsic dimensionality estimate, for the size channel. Within each model, a two-term fit approximates $\mathcal{I}(\beta)$ with high fidelity:

$$\widehat{\mathcal{I}}(\beta) = a \, \log m_k(\beta) + b \, |s(\beta)|. \qquad (12)$$

Within a single model, the spectral slope $|s(\beta)|$ provides a strong empirical one-dimensional proxy for the $\beta$-trajectory of $\mathcal{I}(\beta)$. However, slope alone does not calibrate across architectures with different intrinsic-dimensionality scales. The LID term $m_k(\beta)$ primarily supplies this cross-model normalization, anchoring the proxy scale across backbones and training objectives.

For the coefficient-free selection rule used in the main experiments, we use the product proxy

$$P(\beta) = m_k(\beta) \, |s(\beta)|.$$

This product is a design choice, not a derivation from Eq. 11. It reflects a gating structure: when the spectrum is nearly flat and $|s| \approx 0$, whitening is close to isotropic, so the spectral channel should contribute little regardless of the value of $m_k$. A product enforces this behavior, whereas an additive score does not; ablations are reported in Table 5.

The product $m_k \cdot |s|$ combines two complementary roles. The slope term provides strong within-model tracking of the $\beta$ trajectory, while $m_k$ normalizes across architectures with different intrinsic-dimensionality scales. Importantly, $\mathcal{I}(\beta)$ and $P(\beta)$ are ID-side surrogates: they do not determine FPR by themselves. Their value is empirical and operational, namely that they track the $\beta$ regimes that improve Mahalanobis-style OOD detection in our experiments, without requiring access to OOD samples.

## 8. Conclusion

We studied Mahalanobis-style OOD detection across vision foundation-model backbones, OOD benchmarks, and feature normalizations, finding that performance depends strongly on the representation. A size and stretch factorization links this variation to two ID geometry signals: local intrinsic dimensionality $m_k$ and the within-class spectral slope $|s|$. Within each model, $|s|$ strongly tracks instability along $\beta$ trajectories, while $m_k$ anchors the proxy scale across architectures. Based on this view, we introduced radially scaled $\ell_2$ normalization and an ID-only rule for selecting $\beta$ from $P(\beta) = m_k(\beta)|s(\beta)|$. Under oracle regret, this selection approaches oracle-tuned performance without OOD samples, with the largest gains in near-OOD settings.

## Acknowledgements

This work was supported by the project AITAX (AI Tax Advisor) project under the grant FENG.02.02-IP.05-0314/23, Action 2.2 FIRST TEAM, European Funds for a Modern Economy Programme 2021–2027 (FENG). We gratefully acknowledge the Wrocław Centre for Networking and Supercomputing (WCSS) for providing computational resources and infrastructure.

## Impact Statement

This paper studies how representation geometry and feature normalization affect Mahalanobis-style OOD detection, with the goal of improving the reliability of deployed vision models. The primary positive impact is practical: our findings provide diagnostics and simple post-hoc normalization procedures that can reduce false positives and improve robustness monitoring in safety-relevant settings (e.g., medical imaging, autonomous systems, industrial inspection). Potential risks include misuse for overconfidence: improved OOD detection may be treated as a guarantee of safety, despite the fact that OOD detection remains imperfect and depends on data, model, and deployment conditions. In addition, geometry-based tuning could be adapted to evade certain detectors if an adversary can influence feature distributions. We therefore emphasize that our methods should be used as one component of a broader reliability pipeline (e.g., calibration, auditing, and monitoring), and that deployment decisions should not rely on OOD scores alone.

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

# A. Detailed Description of Spectral and Manifold Metrics

All spectral metrics are computed from the ordered eigenvalues $\lambda_1 \geq \cdots \geq \lambda_d$ of a chosen matrix $M \in \{C, S_w, S_b\}$ (see Appendix C for definitions). When needed, we use the normalized spectrum $p_i \triangleq \lambda_i / \sum_{j=1}^{d} \lambda_j$.

**Intrinsic dimensionality (ID).** A global estimate of manifold dimension using the maximum-likelihood estimator of Ma et al. (2018).

**Local intrinsic dimensionality (LID).** For a feature vector $z$, let $r_j(z)$ denote the distance to its $j$-th nearest neighbor in feature space. The $k$NN LID estimator is

$$\mathrm{LID}_k(z) = -\left[\frac{1}{k} \sum_{j=1}^{k} \log \frac{r_j(z)}{r_k(z)}\right]^{-1}. \tag{13}$$

We report the dataset mean for $k \in \{10, 25, 50, 100\}$.

**Total variance (trace).**

$$\mathrm{Tr}(M) = \sum_{i=1}^{d} \lambda_i. \tag{14}$$

**Effective rank (trace-to-top ratio).** We use the ratio

$$r_{\mathrm{eff}} \triangleq \frac{\sum_{i=1}^{d} \lambda_i}{\lambda_1}. \tag{15}$$

**Participation ratio (PR).**

$$\mathrm{PR} \triangleq \frac{\left(\sum_{i=1}^{d} \lambda_i\right)^2}{\sum_{i=1}^{d} \lambda_i^2}. \tag{16}$$

**Condition number.**

$$\kappa \triangleq \frac{\lambda_1}{\lambda_d}, \tag{17}$$

where $\lambda_d$ is the smallest (non-zero) eigenvalue (or the smallest eigenvalue after numerical regularization, when applicable).

**Spectral gap (head).**

$$\mathrm{Gap} \triangleq \lambda_1 - \lambda_6. \tag{18}$$

**Spectral entropy.**

$$\mathrm{H} \triangleq -\sum_{i=1}^{d} p_i \log p_i, \qquad p_i = \frac{\lambda_i}{\sum_j \lambda_j}. \tag{19}$$

**Average log decay rate (top-20).**

$$\frac{1}{19} \sum_{i=1}^{19} \left(\log \lambda_i - \log \lambda_{i+1}\right). \tag{20}$$

**Log-spectrum slope (full-range).** We fit a least-squares line to the log-spectrum

$$\log \lambda_i = a + b\,i, \tag{21}$$

and report the fitted slope $b$ (in the main paper we often use $|s|$ for the magnitude of this slope when $s < 0$).

**Power-law exponent.** We fit $\lambda_i \propto i^{-\alpha_{\mathrm{PL}}}$ by regressing $\log \lambda_i$ on $\log i$ and report the exponent $\alpha_{\mathrm{PL}}$.

**Dimension for 90% explained variance.**

$$k_{0.9} \triangleq \min \left\{ k : \frac{\sum_{i=1}^{k} \lambda_i}{\sum_{j=1}^{d} \lambda_j} \geq 0.9 \right\}. \tag{22}$$

These definitions match the metrics used in Section 5.2 and enable exact reproducibility.

## B. LID Estimation and Robustness to the Choice of $k$

We estimate local intrinsic dimensionality (LID) using the $k$NN maximum-likelihood estimator of (Ma et al., 2018). For a feature vector $z$, let $r_j(z)$ denote the Euclidean distance to its $j$-th nearest neighbor among ID features (computed in the same feature space as the corresponding experiment, e.g., in $\phi_\beta$-space when sweeping $\beta$). The per-sample estimator is defined in Equation 13. We report the *dataset-average* LID as $m_k \triangleq \frac{1}{N} \sum_{i=1}^{N} \mathrm{LID}_k(z_i)$ on an ID evaluation split (held out from the statistics used to fit Mahalanobis covariances when applicable). Unless stated otherwise, we fix $k = 50$ across all experiments for consistency and denote the resulting estimate by $m \equiv m_{50}$.

**Stability of the product $m|s|$.** Our main-paper summary uses the product $m|s|$, where $|s|$ is the magnitude of the within-class log-spectrum slope of $S_w$ (Section 5.2). Table 6 in Appendix F.4 reports that the proxy tracking behavior remains similar for $k \in \{10, 25, 50, 100\}$, indicating that $m|s|$ is not sensitive to moderate changes in $k$ within the local neighborhood regime used in our experiments.

## C. Computation and Intuition for Covariance and Scatter Matrices

Our spectral analyses are based on three symmetric positive semidefinite matrices computed from in-distribution (ID) feature embeddings. Let $z_i \in \mathbb{R}^d$ be the feature of sample $x_i$ with label $y_i \in \{1, \ldots, K\}$, and let $N$ be the number of ID samples. Define the global mean $\mu \triangleq \frac{1}{N} \sum_{i=1}^{N} z_i$, class means $\mu_k \triangleq \frac{1}{n_k} \sum_{i:y_i=k} z_i$, and class counts $n_k$. All eigenvalues are reported in descending order, $\lambda_1 \geq \cdots \geq \lambda_d$.

**Global covariance ($C$).** We measure the overall spread of ID features with

$$C \triangleq \frac{1}{N} \sum_{i=1}^{N} (z_i - \mu)(z_i - \mu)^\top. \tag{23}$$

Large eigenvalues of $C$ correspond to directions of high variance across the entire ID dataset.

**Within-class scatter ($S_w$).** To measure intra-class variability we use the tied within-class scatter

$$S_w \triangleq \frac{1}{N} \sum_{k=1}^{K} \sum_{i:y_i=k} (z_i - \mu_k)(z_i - \mu_k)^\top. \tag{24}$$

When the Mahalanobis detector is implemented with a shared (tied) covariance across classes, its covariance estimate coincides with $S_w$. In particular, the standard class-conditional MD score can be written as $\mathrm{MD}_k(z) = (z - \mu_k)^\top S_w^{-1} (z - \mu_k)$, matching Eq. 3 in the main text.

**Between-class scatter ($S_b$).** To quantify how class means spread around the global mean, we use

$$S_b \triangleq \frac{1}{N} \sum_{k=1}^{K} n_k (\mu_k - \mu)(\mu_k - \mu)^\top. \tag{25}$$

Large eigenvalues of $S_b$ indicate directions along which class centroids are well separated.

**Spectral Shift Metric.** To study how representations change under distributional shifts, for each matrix $M \in \{C, S_w, S_b\}$ we compute its eigenvalues $\{\lambda_i^{\text{train}}\}$ on the training set and $\{\lambda_i^{\text{eval}}\}$ on a validation or OOD set. The relative eigenvalue shift is defined as

$$\Delta_i(M) = \frac{\lambda_i^{\text{eval}} - \lambda_i^{\text{train}}}{\lambda_i^{\text{train}}}. \tag{26}$$

This spectrum of shifts highlights how the geometry of the representation changes under distributional shift, providing a fine-grained indicator of robustness or overfitting.

**Intuition Behind the Shift Metric.**

- **Zero shift ($\Delta_i \approx 0$):** The corresponding direction in feature space is stable across data splits.

- **Positive shift ($\Delta_i > 0$):** The representation spreads out along this eigenvector in the new data, increasing variance.

- **Negative shift ($\Delta_i < 0$):** The representation compresses along this eigenvector, reducing variance.

- **Magnitude:** Reflects the relative degree of expansion or contraction. For example, $\Delta_i = 0.5$ indicates a 50% increase in variance, while $\Delta_i = -0.2$ indicates a 20% decrease.

**Interpretation in Model Analysis.**

- **Small shifts across all eigenvectors:** Robust and stable representations that generalize well.

- **Large positive shifts:** Features become more variable on new data, potentially indicating under-regularization or sensitivity to OOD inputs.

- **Large negative shifts:** Features compress on new data, potentially indicating overfitting.

- **Consistent shift patterns:** Systematic changes in representation geometry, revealing overfitting or robustness issues.

**Types of Shifts.**

- **Validation covariance shift:** Change in global covariance from training to validation data.

- **OOD covariance shift:** Change in global covariance from training to out-of-distribution data.

- **Validation within-class shift:** Change in within-class scatter from training to validation data.

- **Validation between-class shift:** Change in between-class scatter from training to validation data.

## D. Spectral Analysis of Training Effects

To understand how the intrinsic geometry of representations affects OOD performance, we begin by examining the spectral properties of three key matrices: the feature covariance $C$, the within-class scatter $S_w$, and the between-class scatter $S_b$. These matrices capture complementary aspects of the feature space: $C$ reflects overall variance, $S_w$ measures intra-class dispersion, and $S_b$ quantifies inter-class separation (more details in Appendix C). Our first analysis focuses on the eigenvalue spectra of these matrices. The magnitude and decay of eigenvalues reveal how variance is distributed across dimensions, providing insight into the richness and anisotropy of the feature space. For instance, a steep decay in $S_w$ eigenvalues indicates that intra-class variability is concentrated along a few directions, resulting in tight clusters, whereas a slower decay suggests more diffuse intra-class variation. Similarly, large eigenvalues in $S_b$ correspond to well-separated class means, signaling strong discriminability.

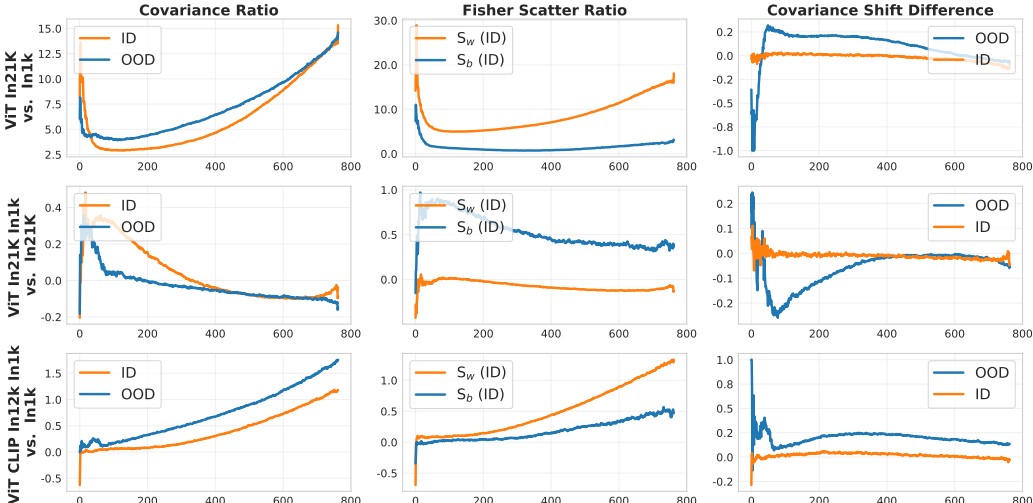

*Figure 10.* Spectral ratios across models. Higher ratios indicate richer within-class variation and more expressive feature spaces. Fine-tuning tends to increase $S_b$ while preserving $S_w$.

**Spectral ratios.** To systematically compare models, we compute ratios between eigenvalues of $S_b$, $S_w$, and $C$. These ratios serve as compact summaries of representation geometry. Higher $S_b/S_w$ ratios indicate representations with greater between-class separation relative to intra-class spread, which generally favors OOD detection, while lower ratios may signal overlapping clusters or limited discriminative power. A higher $C$ ratio indicates that variance is distributed along multiple directions, reflecting a richer and more expressive representation that can better accommodate novel OOD inputs without major distortion. As illustrated in Figure 10, models pretrained on large, diverse datasets (e.g., In21k) exhibit larger $C$ and $S_w$ ratios, capturing richer intra-class variations and producing more expressive feature spaces. Fine-tuning tends to increase $S_b$ ratios while preserving $S_w$, enhancing class separability without sacrificing cluster compactness. Models trained on smaller datasets exhibit smaller ratios, reflecting less expressive representations with weaker discriminability.

**Eigenvalue shifts.** Beyond static spectra, we are interested in how stable the representation geometry is under distributional shifts. To capture this, we define a *spectral shift metric*, which measures the relative change in eigenvalues from the training set to validation or OOD data (see Appendix C). A small shift indicates that the representation preserves its structure across data splits, signaling robustness. Large positive shifts reveal that features are spreading along new directions, while large negative shifts indicate compression. Figure 10 shows that OOD samples induce larger spectral shifts in models trained on small datasets, reflecting lower generalization and brittle feature structures. Large-scale pretrained models show smaller shifts, indicating more stable, robust representations under distributional change. Fine-tuning generally maintains small shifts while increasing $S_b$, improving class separation without compromising intra-class compactness.

## E. Geometry Induced by Radially Scaled $\ell_2$ Normalization and $\beta$ Selection

This appendix provides geometric intuition for the radially scaled $\ell_2$ map $\phi_\beta$ (Sec. 6.2) and specifies the ID-only $\beta$ selection rule used throughout the paper (Sec. 6.4). Our goal is modest: (i) show that $\beta$ continuously reweights *radial* versus *angular* variations, explaining why both LID $m_k(\beta)$ and within-class spectral summaries $s(\beta)$ vary smoothly with $\beta$; and (ii) formalize how we choose $\hat{\beta}$ from the proxy curve $P(\beta)$ without committing a priori to "minimize" or "maximize."

### E.1. Setup and Induced Metric

Let $z \in \mathbb{R}^d \setminus \{0\}$ be a feature vector with Euclidean metric $g_{\text{Euc}}$. Consider the radial map

$$\phi_\beta(z) \;=\; \frac{z}{\|z\|^\beta}, \qquad \beta \in \mathbb{R}, \tag{27}$$

which preserves direction but rescales radius. Writing $z = r\,u$ with $r = \|z\|$ and $u \in S^{d-1}$, the Euclidean metric decomposes as

$$g_{\text{Euc}} \;=\; \mathrm{d}r^2 + r^2 g_{S^{d-1}}. \tag{28}$$

Under $\phi_\beta$, the radius becomes $R = r^{1-\beta}$ and $\mathrm{d}R = (1 - \beta)\, r^{-\beta}\mathrm{d}r$. The pullback metric $g_\beta \triangleq \phi_\beta^* g_{\mathrm{Euc}}$ therefore satisfies

$$g_\beta = (1 - \beta)^2 r^{-2\beta}\, \mathrm{d}r^2 + r^{2(1-\beta)} g_{S^{d-1}}. \tag{29}$$

Eq. 29 makes explicit that $\beta$ changes the relative weighting of radial and angular variations. This is the geometric reason that empirical quantities computed *after* $\phi_\beta$ (e.g., covariance spectra and neighborhood distances used for LID) vary smoothly with $\beta$.

## E.2. How This Interacts with Mahalanobis-Style Scoring

In the main paper we apply $\phi_\beta$ to features and then fit the (tied) Gaussian statistics used by Mahalanobis variants. Equivalently, $\phi_\beta$ changes the distribution of deviations $\delta_\beta(z)$ before inserting them into a quadratic score $S_\beta(z) = \delta_\beta(z)^\top \Sigma(\beta)^{-1}\delta_\beta(z)$. Thus, $\beta$ does not define a new detector; it defines a one-parameter family of *geometrically deformed representations*. Through Eq. 29, this deformation changes: (i) neighborhood structure (hence $m_k(\beta)$), (ii) scatter spectra (hence $s(\beta)$), and (iii) how quadratic weighting amplifies directional deviations.

## E.3. Interpreting $\beta$

Eq. 29 yields a compact interpretation of typical regimes:

- $\beta = 0$ **(identity).** $\phi_\beta$ is the identity and $g_\beta = g_{\mathrm{Euc}}$.

- $0 < \beta < 1$ **(moderate contraction).** Radii shrink as $r^{1-\beta}$ while angular structure is retained.

- $\beta = 1$ **(spherical projection).** Radii become constant ($R = 1$): radial variability is removed and only angles remain.

- $\beta > 1$ **(strong contraction).** Large radii are aggressively compressed; this can suppress variability but may also collapse useful structure.

- $\beta < 0$ **(expansion).** Radii are amplified; norm differences become more prominent, which can increase sensitivity to radial outliers.

## E.4. $\beta$ Selection from the Proxy Curve

For each model we evaluate the proxy

$$P(\beta) = m_k(\beta)\,|s(\beta)| \tag{30}$$

on a discrete grid $\mathcal{B} = \{\beta_1 < \cdots < \beta_T\}$. Empirically, $P(\beta)$ is usually inverted-U shaped (so maximizing $P$ is the typical behavior), but some representations can produce U-shaped curves (where minimizing $P$ is appropriate). Rather than hard-coding "maximize" versus "minimize," we select the most pronounced *interior* turning point.

**Interior turning point rule.** Let $P_t \triangleq P(\beta_t)$. We first define a simple endpoint reference level

$$P_{\mathrm{end}} \triangleq \frac{P_1 + P_T}{2}.$$

Among interior grid points $t \in \{2, \dots, T - 1\}$, we choose the one farthest from this endpoint level:

$$\hat{t} \in \arg\max_{t \in \{2, \dots, T-1\}} |P_t - P_{\mathrm{end}}|, \qquad \hat{\beta} \triangleq \beta_{\hat{t}}. \tag{31}$$

This rule returns the interior *maximum* when the curve is inverted-U, and the interior *minimum* when the curve is U-shaped, while discouraging endpoint-driven choices.

**Nearly monotone curves.** If the proxy has no clear interior turning point on the finite grid (i.e., the largest interior deviation is very small), we fall back to the default behavior and set

$$\hat{\beta} \in \arg\max_{\beta \in \mathcal{B}} P(\beta).$$

In our experiments, this situation is rare and typically occurs when the true extremum lies outside the evaluated range.

**Connection to the main text.** The selection rule is intentionally agnostic to whether $P(\beta)$ should be maximized or minimized. In Sec. 7.2 we further explain why both U-shaped and inverted-U regimes occur by relating the proxy curve to the instability functionals $\mathcal{I}(\beta)$ and its low-dimensional approximation $\widehat{\mathcal{I}}(\beta)$.

### E.5. Proxy Selection Quality vs. Oracle $\beta^*$

To quantify how well the ID-only proxy recovers the best-performing radial exponent, we compare the proxy-selected value $\hat{\beta}$ to the oracle choice $\beta^* \in \arg\min_{\beta \in \mathcal{B}} \mathrm{FPR@95}(\beta)$ computed using OOD labels. For each OOD benchmark, we report the absolute gap in detection performance, $|\mathrm{FPR}(\hat{\beta}) - \mathrm{FPR}(\beta^*)|$, and compare against the fixed normalization baselines $\beta = 0$ (no normalization) and $\beta = 1$ ($\ell_2$ normalization). As shown in Figure 8, proxy selection consistently reduces the oracle gap across datasets and also improves worst-case behavior, indicating that ID geometry carries sufficient signal to guide $\beta$ tuning without access to OOD samples.

## F. Unified Stability Theory: Definitions, Mechanism, and Empirical Evidence

This appendix formalizes the Unified Stability Theory (UST) used in the main text, derives the stretch mechanism, and summarizes supporting empirical evidence. We provide (i) full channel definitions for Eq. 11, (ii) numerical verification of the factorization identity, (iii) intervention experiments isolating the spectrum and allocation effects predicted by Eq. 37, and (iv) proxy ablations and mechanistic evidence for both channels. Unless stated otherwise, all quantities are computed on last-layer features using ID data only.

### F.1. Quadratic Scores and the Instability Functional

Each Mahalanobis-style detector specifies a centred deviation $\delta_\beta(z)$ for feature vector $z$, and a tied scatter estimate $\Sigma(\beta)$ computed from ID data. We assume $\Sigma(\beta) \succ 0$ for all $\beta$ under consideration, ensuring $\Sigma(\beta)^{-1}$ exists, and $\|\delta_\beta(z)\| > 0$ almost surely under $\mathcal{D}_{\mathrm{ID}}$.

The corresponding quadratic score is

$$S_\beta(z) \triangleq \delta_\beta(z)^\top \Sigma(\beta)^{-1} \delta_\beta(z). \tag{32}$$

We quantify ID-side score variability using

$$\mathcal{I}(\beta) \triangleq \mathrm{Var}_{z \sim \mathcal{D}_{\mathrm{ID}}}\big[\log S_\beta(z)\big]. \tag{33}$$

**Factorization and channel decomposition.** For $\delta_\beta(z) \neq 0$, define the *stretch factor*

$$W_\beta(z) \triangleq \frac{\delta_\beta(z)^\top \Sigma(\beta)^{-1} \delta_\beta(z)}{\|\delta_\beta(z)\|^2}, \tag{34}$$

so that $S_\beta(z) = \|\delta_\beta(z)\|^2 W_\beta(z)$ and $\log S_\beta(z) = \log \|\delta_\beta(z)\|^2 + \log W_\beta(z)$. Taking the variance over ID samples gives the exact decomposition

$$\mathcal{I}(\beta) = A_\delta(\beta) + A_W(\beta) + 2A_\times(\beta), \tag{35}$$

where

$$A_\delta(\beta) \triangleq \mathrm{Var}_z\big[\log \|\delta_\beta(z)\|^2\big],$$
$$A_W(\beta) \triangleq \mathrm{Var}_z\big[\log W_\beta(z)\big],$$
$$A_\times(\beta) \triangleq \mathrm{Cov}_z\big(\log \|\delta_\beta(z)\|^2, \log W_\beta(z)\big).$$

In the main text we refer to $A_\delta$ as the *size* channel and $A_W$ as the *stretch* channel.

**Numerical verification.** For each of the 462 (model, $\beta$) configurations we compute $S_\beta$ both directly and via the factored form on 5,000 subsampled ImageNet validation points.

Table 3 confirms the identity holds to floating-point precision across all configurations.

*Table 3.* Numerical residual for the factorization $S_\beta(z) = \|\delta_\beta(z)\|^2 W_\beta(z)$ over all evaluated configurations.

| Statistic | max rel. err | mean rel. err | #configs |
|---|---|---|---|
| All | 7.06e-07 | 4.56e-08 | 462 |

**Empirical channel behaviour.** Both $A_\delta(\beta)$ and $A_W(\beta)$ vary substantially across $\beta$ and models (Figure 11). The cross-term $A_\times$ is negative in 98.3% of (model, $\beta$) pairs (median log-correlation $-0.528$). This sign is geometrically expected: large-norm residuals ($\|\delta\|^2$ large) tend to concentrate along the principal eigen-directions of $\Sigma(\beta)$, which carry the largest eigenvalues $\lambda_i$ and hence the *smallest* whitening weights $1/\lambda_i$, giving a small $W_\beta(z)$. The structural covariance $\mathrm{Cov}(\log\|\delta\|^2, \log W) \approx -0.528$ therefore reflects the whitening operator correctly downweighting well-explained, high-norm residuals. Its practical consequence is that $\mathcal{I}(\beta)$ can remain nearly flat along the $\beta$ trajectory even as OOD performance changes substantially, making $\mathcal{I}$ a less reliable per-model proxy than the individual channels.

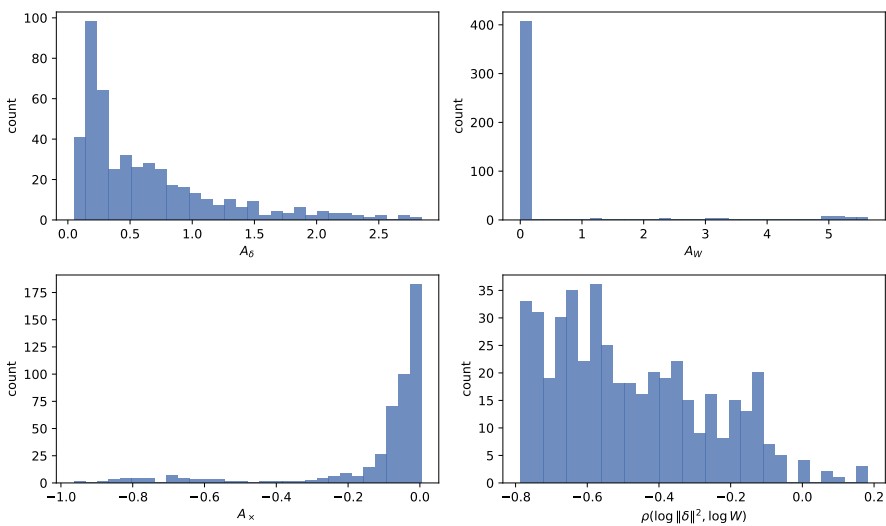

*Figure 11.* Distribution of $A_\delta$, $A_W$, $A_\times$, and $\rho(\log\|\delta\|^2, \log W)$ across all 462 (model, $\beta$) configurations. The cross-term is predominantly negative, consistent with partial cancellation between the size and stretch channels.

## F.2. Stretch Mechanism: Spectral Heterogeneity and Allocation Geometry

Write the eigendecomposition $\Sigma(\beta) = U(\beta)\Lambda(\beta)U(\beta)^\top$ with $\lambda_1(\beta) \geq \cdots \geq \lambda_d(\beta) > 0$. Define the *allocation* of a deviation $\delta_\beta(z)$ onto eigen-directions by

$$p_i(z;\beta) \triangleq \frac{(u_i(\beta)^\top \delta_\beta(z))^2}{\|\delta_\beta(z)\|^2}, \qquad \sum_{i=1}^d p_i(z;\beta) = 1. \tag{36}$$

Substituting into Eq. 34 gives

$$W_\beta(z) = \sum_{i=1}^d \frac{p_i(z;\beta)}{\lambda_i(\beta)}. \tag{37}$$

This identity shows that the stretch factor is jointly controlled by the heterogeneity of whitening weights $\{1/\lambda_i(\beta)\}$ and the sample-dependent allocation $p(z;\beta)$. Variability in $W_\beta$ (and hence $A_W$) can therefore be reduced either by flattening the eigenvalue spectrum or by making allocations more uniform across samples.

## F.3. Intervention Experiments

The following experiments target the mechanisms identified in Eq. 37 individually.

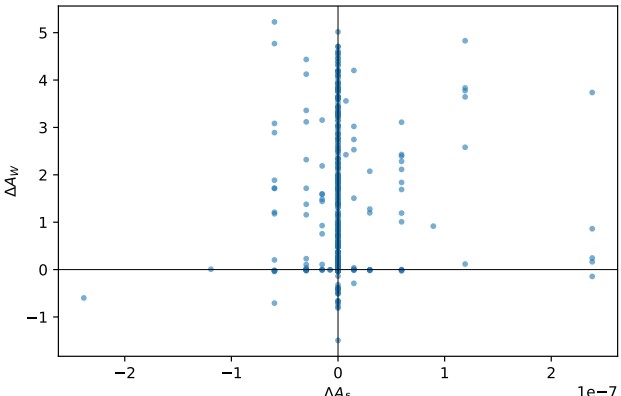

*Figure 12.* Paired $(\Delta A_\delta, \Delta A_W)$ under a fixed random rotation for each (model, $\beta$) configuration. $\Delta A_\delta$ concentrates near zero while $\Delta A_W$ varies substantially, isolating the allocation-geometry factor of the stretch channel.

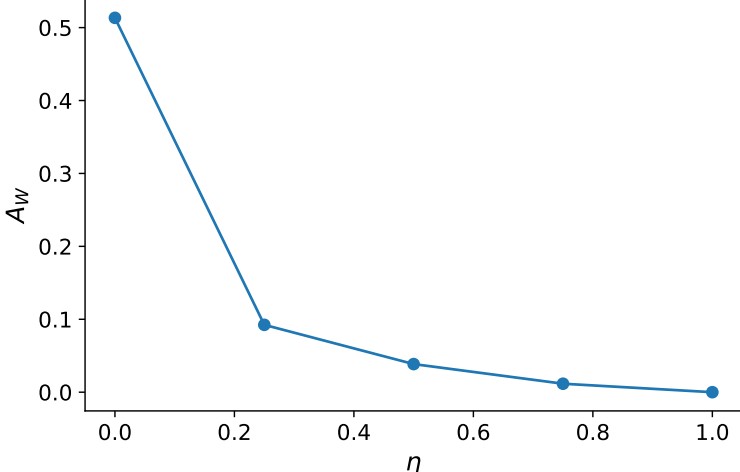

*Figure 13.* $A_W(\eta)$ across models as the allocation is interpolated toward the ID mean. $A_W$ decreases monotonically and collapses to zero at $\eta = 1$.

**Rotation: allocation geometry with norms preserved.** Applying a random orthogonal map $Q \in O(d)$ to the centred residuals, $\delta_\beta(z) \mapsto Q\delta_\beta(z)$, preserves $\|\delta_\beta(z)\|$ but redistributes the projections onto eigen-directions, directly perturbing the allocation $p(z)$. We draw one fixed rotation per model (same across $\beta$) and record $\Delta A_\delta$ and $\Delta A_W$. Across all 462 configurations the mean ratio $|\Delta A_W|/|\Delta A_\delta| = 1.28 \times 10^6$, confirming that $A_W$ encodes alignment with the covariance eigenbasis while $A_\delta$ is unaffected (Figure 12).

**Allocation smoothing: controlled collapse of $A_W$.** We interpolate allocations as $p^{(\eta)}(z) = (1 - \eta)p(z) + \eta\bar{p}, \eta \in [0, 1]$, where $\bar{p}$ is the ID-mean allocation vector. At $\eta = 1$ all samples share the same allocation and $A_W$ collapses to zero. Empirically, $A_W(\eta)$ decreases monotonically for all 22 models (Figure 13), consistent with Eq. 37.

**Radial rescaling: invariance of $A_W$ under fixed $\Sigma$.** Rescaling deviations as $\delta'(z) = c(z)\delta(z)$ for any scalar $c(z) > 0$ leaves

$$W'(z) = \frac{c(z)^2 \, \delta(z)^\top \Sigma^{-1}\delta(z)}{c(z)^2 \, \|\delta(z)\|^2} = W(z),$$

so $A_W$ is invariant when $\Sigma(\beta)$ is held fixed.

Table 4 confirms mean absolute relative change $3.59 \times 10^{-5}$ across 1,386 configurations ($\gamma \in \{0.5, 2.0, 5.0\}$). This

*Table 4.* Scale invariance of $A_W$ under fixed $\Sigma$.

| Metric | mean $|\Delta A_W|$ | #configs |
|---|---|---|
| E8 | 0.000 | 1386 |

*Table 5.* Proxy variant ablation for $\beta^*$ selection (MD target, $k=25$). Normalised gap measures how far the proxy-selected $\beta$ is from the oracle, averaged over models and OOD datasets; lower is better.

| Proxy variant | norm. gap ↓ | accuracy ↑ |
|---|---|---|
| $m_k$ only | 0.207 | 0.793 |
| $|s|$ only | 0.352 | 0.648 |
| $m_k + |s|$ | 0.207 | 0.793 |
| $m_k \times |s|$ | **0.183** | **0.817** |
| $\log m_k + |s|$ | 0.207 | 0.793 |
| $a \log m_k + b|s|$ (LOO) | 0.176 | 0.824 |

invariance implies that size-channel manipulations cannot affect $A_W$ unless the whitening operator $\Sigma^{-1}$ itself changes.

### F.4. From Size and Stretch Channels to the Proxy $P(\beta) = m_k(\beta)\,|s(\beta)|$

This subsection supports the claim in the main text that the ID-only product $P(\beta) = m_k(\beta)\,|s(\beta)|$ captures the variation in $\mathcal{I}(\beta)$ driven by the two UST channels. We organize the evidence into three parts: (A) why the product form, (B) how well the proxy tracks instability along $\beta$, and (C) mechanistic evidence for each channel.

**Spectral slope definition.** Let $\ell_i(\beta) = \log \lambda_i(\beta)$. We fit the affine model $\ell_i \approx \alpha + s \cdot i$ by least squares and take $|s|$ as a scale-free measure of spectral decay (larger $|s|$ corresponds to steeper decay and greater heterogeneity among whitening weights).

**Why the product form.** Within a single model, $|s|$ is a strong empirical for the $\beta$-trajectory of *both* channels: it correlates with $A_W$ at $\rho = 0.971$ and with $A_\delta$ at $\rho = 0.951$ (medians over 22 models, within-model Spearman). This is because power normalisation simultaneously reshapes the eigenvalue spectrum (raising $|s|$) and compresses feature norms (modifying $A_\delta$), so both channels move together as $\beta$ varies. $m_k$ therefore contributes little additional within-model tracking ($\rho(m_k, \mathcal{I}) = 0.481$ vs. $\rho(|s|, \mathcal{I}) = 0.934$); its primary role is *cross-model* normalisation. Additive combinations $m_k + |s|$ collapse to the dominant operand due to scale mismatch, whereas the product integrates both dimensions simultaneously without requiring coefficient fitting. From Eq. 37, spectral heterogeneity also modulates how much allocation geometry matters for $W_\beta$: when $|s| \approx 0$ all whitening weights are equal and residual placement is irrelevant; only when the spectrum is steep does allocation amplify score variability. This cross-channel gating is multiplicative, explaining why the product outperforms both terms independently for cross-model $\beta^*$ selection even though $|s|$ alone dominates within-model tracking.

**Proxy variant ablation.** Table 5 confirms this structure. The product $m_k \cdot |s|$ achieves the lowest normalised gap among coefficient-free variants and matches the leave-one-out fitted linear combination, while additive combinations $m_k + |s|$ perform identically to $m_k$ alone because the sum is dominated by whichever operand has larger scale. Pooled Spearman correlations with FPR tell the same story: the product achieves $\rho = 0.722$ (MD) and $\rho = 0.498$ (RMD), exceeding both individual terms and the additive combination.

**Consistency with the instability decomposition.** To verify that $m_k$ and $|s|$ account for the variation in $\mathcal{I}(\beta)$, we fit the two-term model

$$\widehat{\mathcal{I}}(\beta) = a \log m_k(\beta) + b\,|s(\beta)| \tag{38}$$

within each model by no-intercept least squares. The no-intercept form is appropriate because only the *shape* of the $\beta$-trajectory matters for identifying $\beta^*$; the absolute level of $\mathcal{I}$ is irrelevant.

Table 6 reports within-model Spearman $\rho$ between each proxy variant and $\mathcal{I}(\beta)$ across all 22 models and $k \in \{10, 25, 50, 100\}$. The spectral slope alone achieves $\rho = 0.934$, indicating that within-model $\beta$ trajectories of $\mathcal{I}$ are

*Table 6.* Proxy tracking: within-model Spearman $\rho$ between the proxy and $\mathcal{I}(\beta)$ over $\beta$, across all 22 models. Top block: coefficient-free product $P(\beta) = m_k \cdot |s|$. Bottom block: two-term fitted proxy $\widehat{\mathcal{I}} = a \log m_k + b|s|$ (no curvature; no-intercept OLS).

| Proxy | $k$ | median Spearman $\rho$ | IQR |
|---|---|---|---|
| $m_k \cdot \|s\|$ (coeff-free) | 10 | 0.729 | 0.647 |
| | 25 | 0.751 | 0.701 |
| | 50 | 0.803 | 0.669 |
| | 100 | 0.803 | 0.695 |
| $a \log m_k + b\|s\|$ (fitted, no int.) | 10 | 0.553 | 0.797 |
| | 25 | 0.828 | 0.590 |
| | 50 | 0.918 | 0.602 |
| | 100 | 0.933 | 0.603 |
| $\|s\|$ only | — | 0.934 | 0.602 |

*Table 7.* Within-model Spearman $\rho$ between proxy variants and $\mathcal{I}(\beta)$, summarised as median over 22 models.

| Proxy variant | $k{=}10$ | $k{=}25$ | $k{=}50$ | $k{=}100$ |
|---|---|---|---|---|
| $a \log m_k + b\|s\|$, no intercept | 0.553 | 0.828 | 0.918 | 0.933 |
| $a \log m_k + b\|s\|$, with intercept | 0.909 | 0.823 | 0.839 | 0.810 |
| $m_k \cdot \|s\|$ (coefficient-free) | 0.729 | 0.751 | 0.803 | 0.803 |
| $\|s\|$ only | 0.934 (independent of $k$) | | | |

largely determined by spectral heterogeneity. The two-term fit maintains high fidelity ($\rho = 0.918$ at $k{=}50$; $\rho = 0.933$ at $k{=}100$) and additionally calibrates cross-model variation through $m_k$. The coefficient-free product $m_k \cdot |s|$ achieves $\rho = 0.803$ for $k \geq 50$, capturing the same signal without any regression fitting. Figure 9 visualizes this tracking for three representative models.

**Role of the intercept.** Table 7 examines how the intercept choice affects tracking fidelity. At $k{=}10$ adding an intercept substantially improves performance ($0.553 \to 0.909$), suggesting that at small neighbourhood sizes the no-intercept model underfits the level. At $k \geq 25$ the two variants perform comparably, and we prefer the no-intercept form for theoretical consistency.

**Size–stretch compensation and its effect on $\mathcal{I}$.** As $\beta$ increases, power normalisation compresses feature norms, typically reducing $A_\delta$, while whitening variability grows, increasing $A_W$. The negative cross-term $A_\times$ partially absorbs both, so $\mathcal{I}(\beta)$ can remain nearly flat even as the individual channels—and OOD performance—vary substantially. A natural question is whether removing $A_\times$ (computing $A_\delta + A_W$ directly) reveals $m_k$ as a stronger proxy: it does not. Within-model Spearman correlations between $m_k$ and $\mathcal{I}_{\mathrm{nc}}(\beta) = A_\delta + A_W$ are *lower* than with $\mathcal{I}(\beta)$ ($\rho = 0.466$ vs. $0.481$ at $k{=}25$), confirming that the dominance of $|s|$ arises from its comprehensive coverage of both channels under $\beta$-variation, not from $A_\times$ masking $m_k$. The advantage of $P(\beta) = m_k \cdot |s|$ over $\mathcal{I}(\beta)$ for $\beta^*$ selection is therefore that $|s|$ directly captures the dominant spectral geometry driver for FPR, while $m_k$ adds cross-model normalisation that neither $|s|$ nor $\mathcal{I}$ alone provides. These observations are mutually consistent: when $|s| \approx 0$, $W_\beta \approx 1/\lambda$ for all $z$, so $A_W \approx 0$ and $A_\times \approx 0$, and both the gating argument and the negative-$A_\times$ argument become trivial. In the intermediate regime, the negative $A_\times$ flattens $\mathcal{I}(\beta)$ relative to either channel alone, which is why $P(\beta) = m_k |s|$ outperforms $\mathcal{I}$ for $\beta^*$ selection—it tracks the drivers directly rather than their partially-cancelled sum.

**Stretch channel: interaction moderation.** To verify that spectral heterogeneity gates the influence of $m_k$, we split the 22 models at the median $|s|$ and compare within-model correlations $|\rho(m_k(\beta), \mathrm{FPR}(\beta))|$ in each group (Table 8). Models with steep spectra show substantially stronger $m$–FPR correlation than models with flat spectra for the MD detector, consistent with the prediction from Eq. 37 that only steep spectra make allocation—and hence the geometry captured by $m_k$—performance-relevant.

*Table 8.* Interaction moderation: mean $|\rho(m_k, \mathrm{FPR})|$ within models, split at the median $|s|$ (MD target).

| $k$ | high-$|s|$ group | low-$|s|$ group | moderation $\rho$ |
|-----|-----|-----|-----|
| 10 | 0.802 | 0.585 | 0.360 |
| 25 | 0.751 | 0.574 | 0.359 |
| 50 | 0.690 | 0.568 | 0.269 |

*Table 9.* Size-channel mediation evidence. Panel A reports the within-model Spearman $\rho$ for each link in the chain $m_k \to \sigma_{\mathrm{ID}} \equiv A_\delta \to \mathrm{sep}^{-1} \to \mathrm{FPR}$ (medians over 22 models). Panel B reports raw and partial Spearman $\rho$ between $m_k$ and FPR before and after controlling for score separation $\mathrm{sep} = (\bar{S}_{\mathrm{OOD}} - \bar{S}_{\mathrm{ID}})/\sigma_{\mathrm{ID}}$.

| Panel A: Size-channel chain links | |
|-----|-----|
| **Link** | $\rho$ **(median)** |
| $\sigma_{\mathrm{ID}} \equiv A_\delta$ (within-model $\rho$) | 1.000 |
| $\rho(m_k, \sigma_{\mathrm{ID}})$, $k = 10$ | 0.853 |
| $\rho(m_k, \sigma_{\mathrm{ID}})$, $k = 25$ | 0.868 |
| $\rho(\mathrm{sep}, \mathrm{MD\text{-}FPR})$ | -0.927 |
| $\rho(\mathrm{sep}, \mathrm{RMD\text{-}FPR})$ | -0.930 |

| Panel B: Mediation — $\rho(m_k, \mathrm{FPR})$ before and after controlling for sep | | | | |
|-----|-----|-----|-----|-----|
| Detector | $k$ | $\rho(m_k, \mathrm{FPR})$ | $\rho(m_k, \mathrm{FPR} \mid \mathrm{sep})$ | Drop |
| MD | 10 | 0.891 | 0.564 | 0.327 |
| RMD | 10 | 0.859 | 0.069 | 0.790 |
| MD | 25 | 0.858 | 0.564 | 0.294 |
| RMD | 25 | 0.868 | 0.066 | 0.801 |

**Size channel: score-spread mediation.** To verify that $m_k$'s FPR influence operates through the size channel, we compute per-(model, $\beta$) score statistics directly from raw MD score arrays. Define score separation as $\mathrm{sep} = (\bar{S}_{\mathrm{OOD}} - \bar{S}_{\mathrm{ID}})/\sigma_{\mathrm{ID}}$. Two key empirical observations emerge: (i) the within-ID standard deviation $\sigma_{\mathrm{ID}}$ tracks $A_\delta$ at $\rho = 1.000$ (Spearman) within every model we tested. This near-perfect rank agreement is expected: both quantities are monotone transformations of the dispersion of $\|\delta_\beta\|^2$, so they order (model, $\beta$) configurations identically even though they differ in functional form ($A_\delta$ operates on log-squared norms, $\sigma_{\mathrm{ID}}$ on raw scores); and (ii) $m_k$ predicts $\sigma_{\mathrm{ID}}$ at $\rho = 0.868$ ($k{=}25$; $\rho = 0.853$ at $k{=}10$), because higher intrinsic dimensionality spreads residual norms and hence MD scores.

These links complete the size-channel chain: $m_k \to \sigma_{\mathrm{ID}} \approx_{\mathrm{rank}} A_\delta \to \mathrm{sep}^{-1} \to \mathrm{FPR}$, where $\approx_{\mathrm{rank}}$ denotes rank equivalence across $\beta$ within each model. To confirm mediation, we fit partial Spearman correlations within models. Controlling for score separation reduces $\rho(m_k, \mathrm{RMD\text{-}FPR})$ from 0.87 to 0.07, indicating that the size-channel mechanism fully accounts for $m_k$'s FPR signal for the RMD detector. For MD the residual partial correlation is 0.56, reflecting additional sensitivity to effects not captured by normalised score separation alone. Score separation itself is an excellent direct predictor of FPR (within-model $\rho = -0.927$ for MD, $-0.930$ for RMD). Table 9 reports all chain links and partial-correlation drops for $k \in \{10, 25\}$ and both detectors.

**Spectral slope vs. effective whitening rank.** An alternative spectral summary motivated by the signal-to-noise framing of OOD detection is the *effective rank of the whitening operator*,

$$\mathrm{ER}_w(\beta) = \frac{\left(\sum_i 1/\lambda_i\right)^2}{\sum_i (1/\lambda_i)^2},$$

which quantifies how many eigendirections receive substantial whitening amplification. We test whether $m_k \cdot \mathrm{ER}_w^{-1}$ outperforms $m_k \cdot |s|$ in $\beta^*$ selection using the same evaluation protocol as Table 5. At $k{=}25$ (MD target), $m_k \cdot \mathrm{ER}_w^{-1}$ achieves normalised gap 0.403, versus 0.183 for $m_k \cdot |s|$ and 0.352 for $\mathrm{ER}_w^{-1}$ alone. The spectral slope $|s|$ is decisively the superior spectral summary: it is the log-eigenvalue decay rate, which captures the full gradient of whitening weight heterogeneity, whereas $\mathrm{ER}_w^{-1}$ compresses spectral shape into a single moment ratio that discards ordering information. This confirms that the operative mechanism is spectral *heterogeneity* as characterised by the linear log-eigenvalue slope, not amplification concentration.

$\mathcal{I}(\beta)$ **as a surrogate for FPR.** The instability functional quantifies ID log-score spread; it does not directly determine FPR@95, which also depends on the OOD score distribution. However, for a fixed OOD population, larger ID-side variance pushes the ID tail toward OOD scores, increasing overlap and hence FPR. The decomposition in Eq. 11 therefore identifies *which geometric factors modulate this overlap*, even though the $\mathcal{I}(\beta)$–FPR link varies by detector. Empirically, within-model Spearman correlation between $\mathcal{I}(\beta)$ and FPR@95 is strong for RMD (median $\rho = 0.911$) but moderate for MD (median $\rho = 0.545$). The asymmetry is consistent with UST: MD scores depend on a single quadratic form whose instability $\mathcal{I}$ directly measures, yet FPR also depends on the *mean* ID–OOD gap, which $\mathcal{I}$ does not capture. RMD subtracts a marginal reference that absorbs much of the mean shift, leaving score *variance* as the dominant driver of overlap, hence $\mathcal{I}$ tracks RMD-FPR more faithfully. The product $P(\beta) = m_k \, |s|$ sidesteps this limitation for MD by tracking the channel drivers directly rather than the partially-cancelled sum $\mathcal{I}$.

### F.5. RMD: Term-wise Unified Stability Lens

RMD is commonly implemented as a difference of two Mahalanobis-type quadratic forms, $S_{\mathrm{RMD},\beta}(z) = S_\beta^{(1)}(z) - S_\beta^{(2)}(z)$, where the deviations $\delta_\beta^{(j)}$ and scatter estimates $\Sigma^{(j)}(\beta)$ may differ between terms. As a result, RMD does not generally admit the single-$\Sigma$ representation in Eq. 7.

**Why term-wise.** Eq. 7 in the main text covers detectors whose score is a *single* quadratic form (e.g., MD, MMD). RMD is a *difference* of two quadratic forms that generally use distinct scatter estimates, so it does not admit a single shared $\Sigma(\beta)$ in general. UST therefore applies to RMD *term-wise*: each quadratic component admits the same size–stretch factorization and channel decomposition, and RMD behavior can additionally depend on interactions between the two terms (e.g., cancelation).

### F.6. Exact Term-Wise Factorization and Channels

For each term $j \in \{1, 2\}$, define

$$S_\beta^{(j)}(z) = \delta_\beta^{(j)}(z)^\top \Sigma^{(j)}(\beta)^{-1} \delta_\beta^{(j)}(z), \qquad W_\beta^{(j)}(z) = \frac{S_\beta^{(j)}(z)}{\|\delta_\beta^{(j)}(z)\|^2}.$$

Then the exact factorization holds term-wise:

$$S_\beta^{(j)}(z) = \|\delta_\beta^{(j)}(z)\|^2 \, W_\beta^{(j)}(z), \qquad \log S_\beta^{(j)}(z) = \log \|\delta_\beta^{(j)}(z)\|^2 + \log W_\beta^{(j)}(z),$$

and the instability functional for each term decomposes as

$$\mathcal{I}^{(j)}(\beta) \triangleq \mathrm{Var}[\log S_\beta^{(j)}] = A_\delta^{(j)}(\beta) + A_W^{(j)}(\beta) + 2A_\times^{(j)}(\beta),$$

with the obvious definitions of $A_\delta^{(j)}, A_W^{(j)}, A_\times^{(j)}$.

# G. Detailed OOD Performance

Table 10 reports FPR@95 (↓) for each (model, detector) pair together with the corresponding validation accuracy. Across a wide range of backbones, the radially scaled (RS) variants provide the most reliable gains within the Mahalanobis families: whenever RS-MD or RS-RMD is compared against its direct baselines (MD++/MD or RMD++/RMD), it frequently attains a lower FPR@95, indicating that radially scaled normalization is a strong and generally beneficial modification. Although a few models still achieve their absolute best score with non-Mahalanobis detectors (e.g., MLS or VIM in isolated cases), the RS variants remain consistently competitive and, in aggregate, deliver the lowest mean FPR@95 across models. This pattern suggests that the RS mechanism improves robustness across heterogeneous pretrained representations, whereas accuracy alone is not predictive of OOD performance; high-accuracy models can still exhibit higher FPR@95 than lower-accuracy ones, motivating direct evaluation of OOD detection rather than using accuracy as a proxy.

*Table 10.* **FPR@95 (↓) and validation accuracy (Acc, %) across models and detectors.** Bold denotes the lowest FPR@95 *within each model* across all detectors. Light green highlights cases where RS-MD outperforms both MD++ and MD, and where RS-RMD outperforms both RMD++ and RMD; in these cases, the corresponding baselines are grayed out. The Avg row reports the mean FPR@95 across models for each detector.

| Model | Acc | MSP | MLS | KNN | VIM | RS-MD | MD++ | MD | RS-RMD | RMD++ | RMD |
|---|---|---|---|---|---|---|---|---|---|---|---|
| BEiTV2 FT In1k | 85.5 | 52.2 | 50.7 | 42.6 | 39.3 | 37.2 | 37.6 | 40.2 | **37.1** | 37.3 | 39.1 |
| BEiTV2 FT In21k | 85.1 | 38.0 | **25.8** | 35.1 | 29.5 | 29.8 | 29.8 | 43.6 | 32.5 | 32.5 | 33.1 |
| DINOV2 | 83.0 | 44.9 | 32.8 | 40.2 | **28.9** | 33.8 | 34.5 | 33.8 | 41.4 | 41.2 | 41.4 |
| MAE FT In1k | 83.5 | 54.2 | 55.7 | 44.4 | 40.6 | 39.9 | 40.3 | 43.5 | **38.9** | 39.3 | 41.7 |
| ViT | 77.1 | 56.5 | 50.4 | 50.0 | 53.0 | 45.5 | 45.4 | 45.7 | 44.8 | **44.6** | 44.9 |
| ViT-S In21K In1k | 75.8 | 57.2 | 44.4 | 53.8 | **41.2** | 42.6 | 41.6 | 51.0 | 45.4 | 45.2 | 44.9 |
| ViT In21K In1k | 78.5 | 53.7 | 40.7 | 47.7 | 36.1 | 35.8 | 38.7 | **35.7** | 37.5 | 37.6 | 37.6 |
| ViT-L In21K In1k | 83.6 | 44.8 | 29.8 | 34.3 | **25.0** | 25.3 | 28.2 | 25.3 | 26.9 | 26.9 | 26.9 |
| ViT CLIP In1k | 84.7 | 55.2 | 65.3 | 41.1 | 41.7 | **37.6** | 38.2 | 40.2 | 38.2 | 38.6 | 40.3 |
| ViT CLIP In12k In1k | 85.4 | 49.0 | 51.9 | 32.5 | 30.0 | **26.4** | 27.8 | 33.5 | 30.7 | 30.9 | 32.5 |
| ViT-L CLIP In12k In1k | 86.1 | 45.0 | 43.6 | 30.1 | 28.0 | **26.7** | 27.1 | 29.7 | 27.0 | 27.7 | 29.3 |
| EVA02 | 82.0 | 49.1 | 39.5 | 54.7 | **38.8** | 40.6 | 44.6 | 53.4 | 44.0 | 44.0 | 44.0 |
| EVA02 FT In1k | 84.2 | 53.2 | 55.3 | 40.6 | 43.9 | **37.2** | 37.4 | 37.6 | 39.6 | 39.8 | 40.3 |
| EVA02 FT In21k | 80.2 | 44.6 | 34.3 | 55.0 | **32.9** | 50.6 | 50.6 | 56.5 | 37.7 | 37.7 | 36.7 |
| EVA02-S FT In22k In1k | 82.2 | 59.2 | 64.8 | 44.2 | 44.8 | **40.8** | 41.0 | 42.2 | 43.3 | 43.6 | 44.3 |
| EVA02 FT In21k In1k | 82.2 | 53.0 | 58.9 | 42.3 | **37.1** | 39.5 | 38.2 | 40.8 | 38.6 | 39.1 | 40.3 |
| EVA02-L FT In22k In1k | 84.8 | 43.8 | 43.0 | 38.4 | 36.9 | 40.2 | 37.8 | 37.0 | **31.6** | 31.8 | 32.6 |
| DeiT3 | 83.5 | 55.0 | 59.2 | 47.5 | 47.2 | 43.2 | 43.0 | 43.3 | **39.4** | 39.9 | 40.8 |
| DeiT3 In21k In1k | 85.0 | 56.7 | 64.3 | 37.0 | 37.5 | 35.5 | 35.6 | 37.6 | **34.5** | 35.1 | 36.6 |
| DeiT3-L In22k In1k | 85.7 | 58.1 | 65.9 | 35.9 | 39.7 | **33.4** | 34.2 | 36.6 | 35.5 | 35.9 | 37.6 |
| DeiT3 FB In22k In1k | 83.8 | 60.9 | 64.9 | 41.1 | 39.8 | 40.2 | **38.8** | 40.7 | 39.2 | 39.2 | 40.5 |
| Avg | | 51.6 | 49.6 | 42.3 | 37.7 | **37.2** | 37.6 | 40.4 | **37.3** | 37.5 | 38.3 |

*Table 11.* **FPR on NINCO across model families for Mahalanobis variants** (lower is better). **MD\*** uses the empirically optimal $\beta$; $\hat{\text{MD}}$ uses the regression-predicted $\hat{\beta}$; **MD** (standard) fixes $\beta = 0$; and **MD++** (Mahalanobis++) fixes $\beta = 1$.

| Model | Acc | MSP | MLS | KNN | VIM | RS-MD | MD++ | MD | RS-RMD | RMD++ | RMD |
|---|---|---|---|---|---|---|---|---|---|---|---|
| BEiTV2 FT In1k | 85.5 | 57.0 | 63.9 | 56.6 | 55.1 | 46.3 | 47.0 | 50.6 | **43.9** | 44.3 | 45.6 |
| BEiTV2 FT In21k | 85.1 | 40.8 | **28.7** | 44.3 | 33.0 | 36.4 | 36.4 | 47.5 | 33.1 | 33.1 | 33.9 |
| DINOV2 | 83.0 | 48.3 | 37.2 | 54.7 | **35.2** | 42.4 | 44.5 | 42.4 | 57.7 | 57.4 | 57.7 |
| MAE FT In1k | 83.5 | 56.4 | 67.0 | 51.9 | 51.8 | 48.5 | 48.8 | 51.6 | **44.6** | 45.0 | 46.3 |
| ViT | 77.1 | 61.8 | 63.5 | 58.7 | 71.4 | 55.6 | 55.6 | 55.7 | 51.4 | **51.4** | 51.6 |
| ViT-S In21K In1k | 75.8 | 60.3 | 54.1 | 60.1 | 52.0 | 53.1 | **50.5** | 51.5 | 51.3 | 51.4 | 51.2 |
| ViT In21K In1k | 78.5 | 61.1 | 49.3 | 54.6 | 46.6 | 41.7 | 48.1 | **40.5** | 42.2 | 42.5 | 42.5 |
| ViT-L In21K In1k | 83.6 | 45.9 | 34.0 | 41.5 | 30.9 | 31.0 | 35.8 | 32.2 | **26.9** | 27.7 | 27.2 |
| ViT CLIP In1k | 84.7 | 59.2 | 79.4 | 46.6 | 54.9 | 44.8 | 45.5 | 47.6 | **42.8** | 43.3 | 44.6 |
| ViT CLIP In12k In1k | 85.4 | 49.2 | 68.5 | 39.1 | 35.4 | **31.7** | 33.6 | 37.9 | 34.5 | 35.2 | 37.3 |
| ViT-L CLIP In12k In1k | 86.1 | 45.4 | 59.1 | 33.7 | 32.9 | 33.5 | 31.0 | 32.0 | **28.4** | 29.0 | 29.9 |
| EVA02 | 82.0 | 54.7 | **50.5** | 63.2 | 58.7 | 59.0 | 63.0 | 69.1 | 52.8 | 52.3 | 51.5 |
| EVA02 FT In1k | 84.2 | 58.8 | 75.5 | 46.8 | 71.0 | **41.0** | 41.3 | 41.8 | 42.2 | 42.5 | 43.1 |
| EVA02 FT In21k | 80.2 | 47.1 | 45.8 | 56.5 | 40.7 | 56.3 | 56.3 | 63.8 | 40.0 | 40.0 | **39.3** |
| EVA02-S FT In22k In1k | 82.2 | 67.2 | 82.5 | 54.5 | 69.4 | **49.9** | 50.4 | 52.6 | 51.2 | 51.3 | 51.7 |
| EVA02 FT In21k In1k | 82.2 | 57.2 | 82.7 | 46.1 | 50.9 | 43.4 | 40.7 | 42.4 | **40.5** | 40.6 | 42.5 |
| EVA02-L FT In22k In1k | 84.8 | 48.0 | 68.4 | 37.0 | 59.5 | 41.7 | 38.8 | 35.6 | **31.5** | 31.6 | 32.1 |
| DeiT3 | 83.5 | 58.5 | 70.4 | 55.8 | 63.6 | 49.9 | 50.0 | 50.5 | **43.2** | 43.3 | 43.7 |
| DeiT3 In21k In1k | 85.0 | 64.9 | 86.3 | 44.8 | 54.1 | 42.6 | 42.3 | 43.2 | **37.8** | 38.7 | 40.7 |
| DeiT3-L In22k In1k | 85.7 | 65.2 | 84.8 | 40.1 | 55.8 | 39.0 | 38.8 | 40.2 | **38.1** | 38.5 | 40.5 |
| DeiT3 FB In22k In1k | 83.8 | 63.8 | 82.0 | 47.7 | 57.8 | 47.8 | 45.5 | 48.0 | 44.5 | **44.5** | 45.7 |
| Avg | | 55.8 | 63.5 | 49.3 | 51.5 | **44.5** | 45.0 | 46.5 | **41.8** | 42.1 | 42.8 |

# H. Oracle-Regret Analysis of the Proxy Selector

This appendix provides the supporting regret analyses summarized in Section 6.4.

**Default protocol.** Unless noted, all regret values are computed per model–dataset pair on the full $\beta$ grid $\mathcal{B} = \{-2, -1.75, \ldots, 2\}$ (step 0.25), then averaged across the foundation-model backbones (full list in Appendix L) and the five OOD benchmarks (NINCO, SSB-Hard, OpenImages-O, Textures, iNaturalist). Two subsections use a restricted protocol and flag it explicitly: Section H.1 reports per-dataset breakdowns (no dataset averaging); Section H.6 uses a coarsened $\beta$ grid and model-averaged FPR curves for tractability. Regret is reported in FPR@95 percentage points; Near@1% is the fraction of cases where the selected $\hat{\beta}$ falls within 1 pp of the oracle $\beta^*$.

## H.1. Per-Dataset Regret vs Fixed $\beta = 1$

Proxy selection consistently reduces oracle regret relative to fixed $\beta = 1$ on every dataset for MD. The gains are most pronounced on harder near-OOD benchmarks: on SSB-Hard, MeanReg drops from 4.41 to 3.16 for MD. The worst-case reduction on iNaturalist for MD (MaxReg: $25.39 \rightarrow 14.72$) illustrates that the proxy avoids the catastrophic mistuning a fixed choice can produce.

Table 12. Per-dataset oracle regret: proxy vs fixed $\beta = 1$. Regret in FPR@95 percentage points.

| Dataset | Selector | MeanReg | P90Reg | MaxReg |
|---|---|---|---|---|
| NINCO | $\beta = 1$ | 2.56 | 6.32 | 8.75 |
|  | **Proxy** | **1.92** | **3.99** | **8.32** |
| SSB-Hard | $\beta = 1$ | 4.41 | 12.10 | 16.24 |
|  | **Proxy** | **3.16** | **7.34** | **12.01** |
| OpenImages-O | $\beta = 1$ | 1.10 | 2.38 | 3.83 |
|  | **Proxy** | **0.73** | **1.68** | **2.77** |
| Textures | $\beta = 1$ | 2.39 | 5.25 | 6.78 |
|  | **Proxy** | **1.92** | **4.29** | **5.17** |
| iNaturalist | $\beta = 1$ | 3.34 | 8.42 | 25.39 |
|  | **Proxy** | **2.24** | **6.21** | **14.72** |

## H.2. Near-OOD vs Far-OOD Split

Grouping the five benchmarks into near-OOD (NINCO, SSB-Hard) and far-OOD (iNaturalist, OpenImages-O, Textures) reveals that the proxy's advantage is *larger* in the near-OOD regime—precisely where tuning matters most. Aggregating across MD and RMD, proxy selection reduces MeanReg from 1.98 to 1.27, P90Reg from 5.46 to 2.77, and MaxReg from 16.24 to 9.03 on near-OOD; far-OOD improvements are smaller in absolute terms because $\beta = 1$ already performs well there.

Table 13. Oracle regret split by OOD difficulty. Near-OOD: NINCO, SSB-Hard. Far-OOD: iNaturalist, OpenImages-O, Textures. Aggregated across MD and RMD.

| OOD split | Selector | MeanReg | P90Reg | MaxReg |
|---|---|---|---|---|
| Near-OOD | $\beta = 1$ | 1.98 | 5.46 | 16.24 |
|  | **Proxy** | **1.27** | **2.77** | **9.03** |
| Far-OOD | $\beta = 1$ | 1.16 | 2.65 | 6.49 |
|  | **Proxy** | **0.80** | **2.05** | **4.94** |

The near-OOD MaxReg reduction ($16.24 \rightarrow 9.03$) shows the selector avoids catastrophic mistuning in the hardest cases. A finer analysis of $\hat{\beta}$ distributions across near/far splits is left for future work, but these results indicate the selector remains stable in the practically important near-OOD setting and that its relative benefit increases there.

### H.3. Robustness to LID Neighborhood Size $k$

The proxy $m \cdot |s|$ is not brittle to the choice of $k$. For MD, MeanReg varies only from $1.99$ to $2.75$ across $k \in \{10, 25, 50, 100\}$, with P90Reg tightly clustered (5.38–5.75). For RMD, performance improves steadily with $k$, with Near@1% rising from 53.33 to 72.22. We adopt $k = 50$ throughout the main paper as a balance between MD and RMD performance.

*Table 14.* Oracle-regret robustness to LID neighborhood size $k$. $|\rho|$ is the pooled Spearman correlation between $P(\beta) = m_k|s|$ and FPR@95.

| Detector | $k$ | MeanReg | P90Reg | Near@1% | $|\rho|$ |
|---|---|---|---|---|---|
| MD | 10 | 2.28 | 5.50 | 40.00 | 0.72 |
| | 25 | 2.04 | 5.38 | 48.89 | 0.72 |
| | 50 | 1.99 | 5.44 | 46.67 | 0.69 |
| | 100 | 2.75 | 5.75 | 43.33 | 0.72 |
| RMD | 10 | 1.17 | 2.46 | 53.33 | 0.66 |
| | 25 | 1.26 | 2.63 | 48.89 | 0.64 |
| | 50 | 0.93 | 2.30 | 64.44 | 0.68 |
| | 100 | 0.82 | 2.14 | 72.22 | 0.69 |

### H.4. Proxy vs Alternative Geometric Selectors

We compare $m \cdot |s|$ against five alternative coefficient-free selectors derived from the same ID features. The proxy achieves the best oracle regret on MeanReg, P90Reg, and matches the best on Near@1%.

*Table 15.* Proxy vs alternative geometric selectors (MD target, $k = 50$ for the proxy). RMD orderings are consistent with MD (full RMD breakdown omitted for space).

| Selector | MeanReg | P90Reg | Near@1% | $|\rho|$ |
|---|---|---|---|---|
| **Proxy ($m \times |s|$)** | **1.99** | **5.44** | 46.67 | 0.69 |
| LID ($k$=10) | 2.65 | 7.01 | 44.44 | 0.70 |
| LID ($k$=25) | 2.67 | 7.51 | 50.00 | 0.74 |
| LID ($k$=50) | 2.72 | 6.44 | 37.78 | 0.72 |
| LID ($k$=100) | 2.35 | 6.93 | 48.89 | 0.67 |
| Mean Deviation | 3.58 | 7.91 | 48.89 | 0.67 |
| Fisher Ratio | 4.90 | 17.89 | 46.67 | 0.73 |
| Effective Rank | 5.64 | 18.03 | 33.33 | 0.67 |
| Entropy | 6.63 | 18.67 | 30.00 | 0.66 |

Notably, Mean Deviation—which Mahalanobis++ (Mueller & Hein, 2025) uses to motivate unit-sphere normalization—is not the best statistic for *selecting* the optimal $\beta$. Even the strongest standalone LID variant (LID $k$=100) underperforms the proposed product proxy. This supports the gating-structure argument of Section 7.2: neither factor alone captures the within-model $\beta$-trajectory, but their product does.

### H.5. Turning-Point Rule vs Naive Max/Min

The turning-point rule (Section 6.4, Appendix E.4) is a heuristic; we evaluate it against naive extremum extraction of $P(\beta)$ over the full $\beta$ grid. Selecting $\hat{\beta} = \arg\max_\beta P(\beta)$ over the grid yields MeanReg $= 1.99$ and MaxReg $= 24.08$; selecting $\hat{\beta} = \arg\min_\beta P(\beta)$ is uniformly worse. The interior turning-point rule reduces MeanReg by more than $2\times$ relative to argmax and reduces MaxReg by nearly $3\times$ (9.03 vs. 24.08).

This confirms the rule's practical value: it is more robust than either naive alternative when proxy-curve shape varies across architectures (some models exhibit inverted-U curves, others U-shaped—see Appendix E.4). The turning-point rule is a geometrically motivated heuristic, not a theorem, but it demonstrably yields low-regret ID-only tuning.

*Table 16.* Turning-point rule vs. naive extremum extraction over the full $\beta$ grid (MD target, $k = 50$, aggregated across all backbones and OOD datasets).

| Rule | MeanReg | P90Reg | MaxReg | Near@1% |
|------|---------|--------|--------|---------|
| **Turning-point** | **0.89** | **2.20** | **9.03** | **74.17** |
| $\arg\max_\beta P(\beta)$ | 1.99 | 4.03 | 24.08 | 65.83 |
| $\arg\min_\beta P(\beta)$ | 5.10 | 11.97 | 26.15 | 23.33 |

### H.6. Decomposition-Based Selector Comparison

We test whether using the full instability decomposition Eq. (11) directly as a selector would outperform the simpler proxy. On a coarsened $\beta$ grid ($\beta \in \{-1, -0.5, 0, 0.5, 1, 1.5, 2\}$) with model-averaged FPR curves—a protocol chosen for tractability of the full decomposition fit—the proxy outperforms both the full decomposition and the decomposition without the compensation term on every metric.

*Table 17.* Selector ablation against decomposition-based alternatives, on a coarsened $\beta$ grid with model-averaged FPR curves. Note that the absolute regret values are not comparable to Table 14 due to the protocol change.

| Selector | MeanReg | P90Reg | MaxReg | Near@1% | $|\rho|$ |
|----------|---------|--------|--------|---------|-----------|
| **Proxy** | **1.47** | **4.32** | **8.32** | 61.11 | **0.71** |
| No compensation ($A_\delta + A_W$) | 2.23 | 6.59 | 13.83 | 61.11 | 0.54 |
| Full $\mathcal{I}_{\text{total}}$ | 3.75 | 9.88 | 15.67 | 44.44 | 0.54 |

Dropping the compensation term improves selection (confirming it introduces noise into the explanatory identity), but the simpler proxy remains superior. This supports the intended interpretation in Section 7.2: the decomposition is most useful as an explanatory identity of the relevant geometric channels, while the product proxy is the practical selector designed for robust ID-only tuning.

## I. Geometry of Eigenvalues

Figures 14–19 provide the full eigenspectrum and covariance-shift plots supporting the spectral summaries in Figure 10.

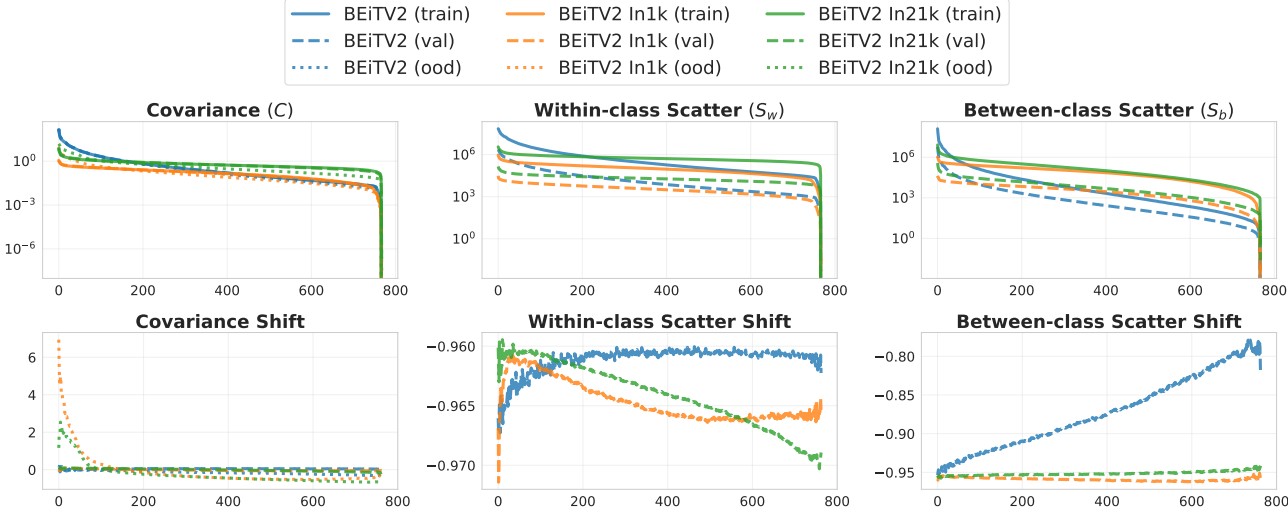

*Figure 14.* BEiTV2 eigenspectra and their respective shifts: top—eigenvalues of covariance C, within-class $S_w$, and between-class $S_b$ across train (solid), val (dashed), and OOD (dotted); bottom—corresponding OOD-induced eigenvalue shifts relative to train.

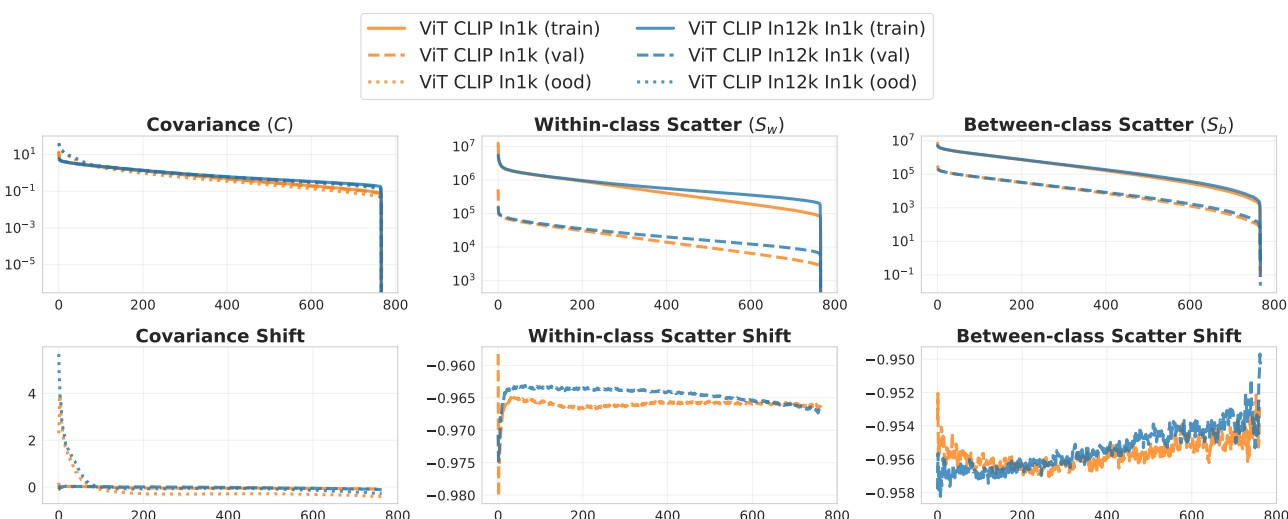

*Figure 15.* CLIP eigenspectra and their respective shifts: top—eigenvalues of covariance C, within-class $S_w$, and between-class $S_b$ across train (solid), val (dashed), and OOD (dotted); bottom—corresponding OOD-induced eigenvalue shifts relative to train.

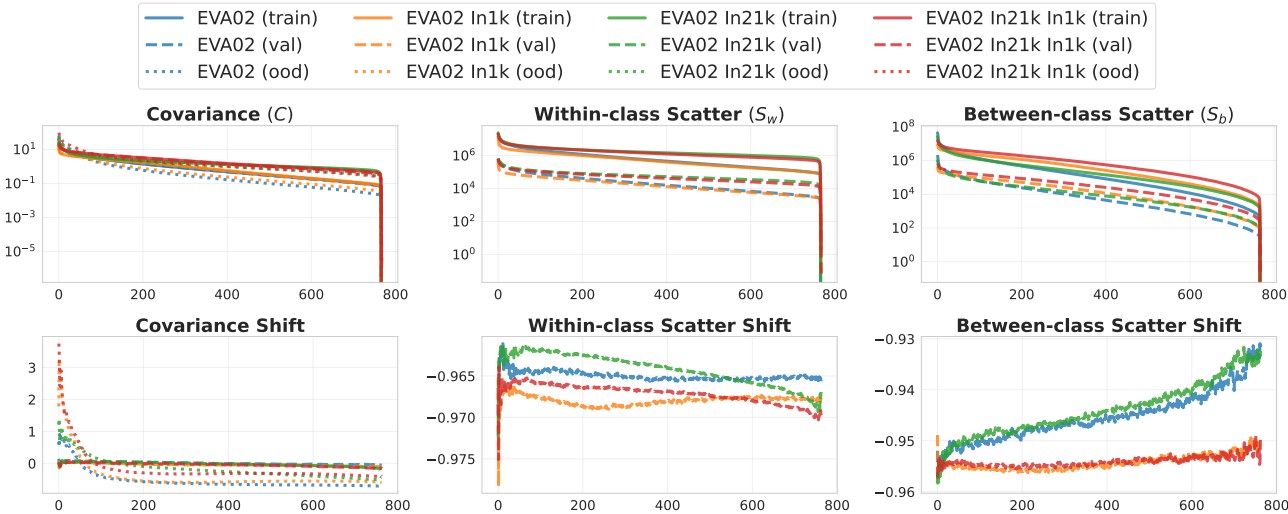

*Figure 16.* EVA02 eigenspectra and their respective shifts: top—eigenvalues of covariance C, within-class $S_w$, and between-class $S_b$ across train (solid), val (dashed), and OOD (dotted); bottom—corresponding OOD-induced eigenvalue shifts relative to train.

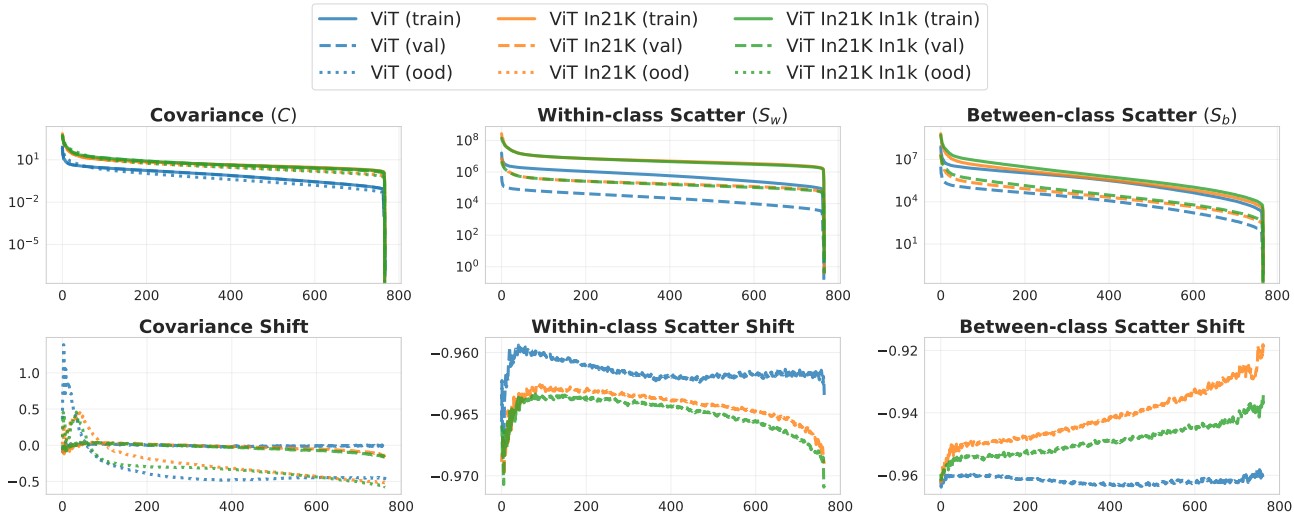

*Figure 17.* ViT eigenspectra and their respective shifts: top—eigenvalues of covariance C, within-class $S_w$, and between-class $S_b$ across train (solid), val (dashed), and OOD (dotted); bottom—corresponding OOD-induced eigenvalue shifts relative to train.

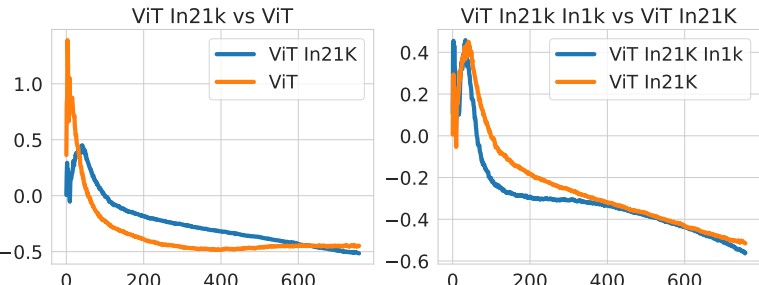

*Figure 18.* Eigenspectrum of covariance shift between train and OOD data (NINCO) for ViT variants: left—ViT In21K vs ViT; right—ViT In21K In1k vs ViT In21K.

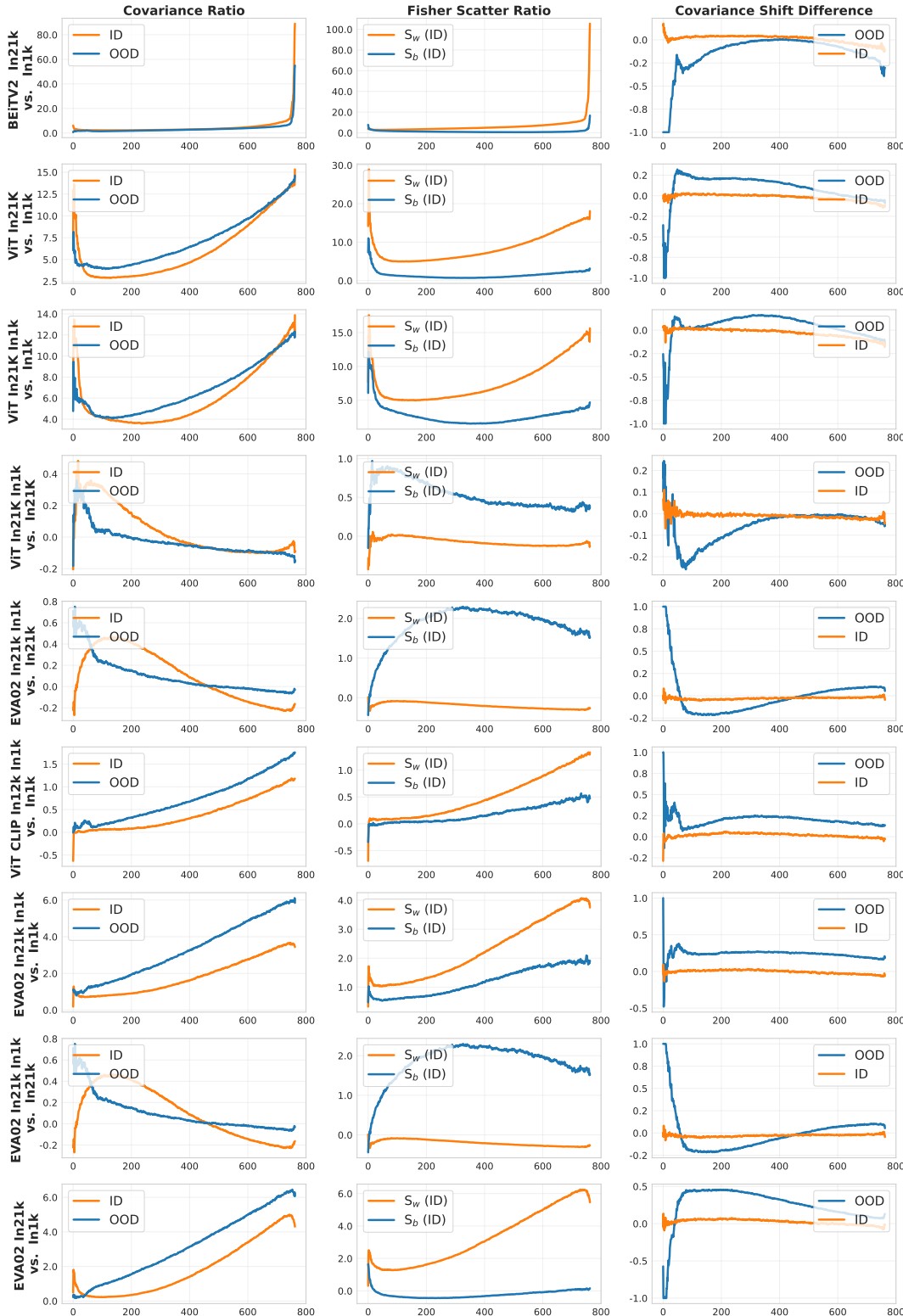

*Figure 19.* Eigenspectrum differences by model pair (BEiTV2, ViT, EVA02/CLIP): for each pair, we plot ID vs OOD covariance $C$, ID within-class $S_w$ and between-class $S_b$, and covariance-shift curves (OOD and ID), showing relative eigenvalue changes between the first and second model.

## J. Pearson Correlations

We replicate the correlation analysis from the main paper using Pearson correlations instead of Spearman correlations (Figure 20). The trends remain consistent: manifold-geometry and eigenvalue-based metrics show similar relationships with OOD performance across the three Mahalanobis variants, confirming that the observed patterns are not sensitive to the choice of correlation metric.

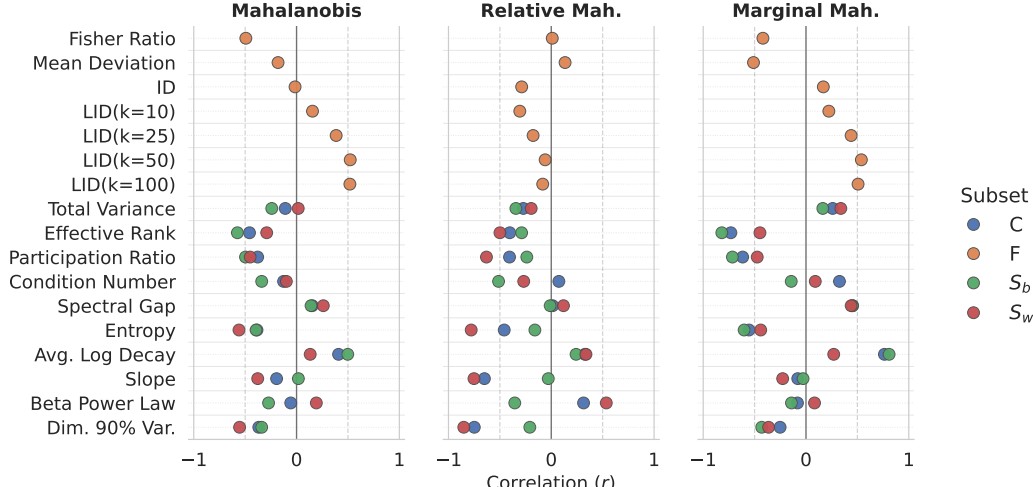

*Figure 20.* Pearson correlations between representation metrics and OOD performance across Mahalanobis variants. The three Mahalanobis-based detectors exploit different geometric cues, leading to distinct correlation patterns consistent with the Spearman results in Figure 4.

## K. Evaluation Protocol and Models

**Models** We evaluate publicly available checkpoints from `timm` (Wightman, 2019) and `huggingface-transformers` (Wolf et al., 2020), covering multiple transformer families, model scales, and training objectives. The full model list is provided in Appendix L.

**Evaluation protocol** Following OpenOOD (Yang et al., 2022), ImageNet-1K serves as the in-distribution (ID) dataset (train features for fitting; validation for ID testing). We report FPR@95 for distinguishing ImageNet validation from each of the five OOD benchmarks: NINCO (Bitterwolf et al., 2023), iNaturalist (Van Horn et al., 2018), SSB-Hard (Bitterwolf et al., 2023), OpenImages-O (Krasin et al., 2017), and Textures (Cimpoi et al., 2014). Unless stated otherwise, we fit class means $\{\mu_k\}$ and a tied covariance $\Sigma$ on ImageNet-1K training features and evaluate OOD scores on the ImageNet validation set versus each OOD dataset.

## L. Full Model Names

*Table 18.* Mapping of model names to checkpoints and sources.

| Model Name | Checkpoint (Version) | Source |
| --- | --- | --- |
| BEiTV2 In1k | beitv2_base_patch16_224.in1k_ft_in1k | timm / huggingface |
| BEiTV2 In21k | beitv2_base_patch16_224.in1k_ft_in22k | timm / huggingface |
| DINOV2 | vit_base_patch14_dinov2.lvd142m | timm / huggingface |
| MAE In1k | mae_finetuned_vit_base | github.com/facebookresearch/mae |
| ViT | vit_base_patch16_224.augreg_in1k | timm / huggingface |
| ViT In21K | vit_base_patch16_224.augreg_in21k | timm / huggingface |
| ViT In21K In1k | vit_base_patch16_224.augreg_in21k_ft_in1k | timm / huggingface |
| ViT-S In21K In1k | vit_small_patch16_224.augreg_in21k_ft_in1k | timm / huggingface |
| ViT-L In21K In1k | vit_large_patch16_224.augreg_in21k_ft_in1k | timm / huggingface |
| ViT CLIP In1k | vit_base_patch16_clip_224.laion2b_ft_in1k | timm / huggingface |
| ViT CLIP In12k In1k | vit_base_patch16_clip_224.laion2b_ft_in12k_in1k | timm / huggingface |
| ViT-L CLIP In12k In1k | vit_large_patch14_clip_336.laion2b_ft_in12k_in1k | timm / huggingface |
| EVA02 | eva02_base_patch14_224.mim_in22k | timm / huggingface |
| EVA02 In1k | eva02_base_patch14_448.mim_in22k_ft_in1k | timm / huggingface |
| EVA02 In21k | eva02_base_patch14_448.mim_in22k_ft_in22k | timm / huggingface |
| EVA02 In21k In1k | eva02_base_patch14_448.mim_in22k_ft_in22k_in1k | timm / huggingface |
| EVA02-L In22k In1k | eva02_large_patch14_448.mim_m38m_ft_in22k_in1k | timm / huggingface |
| EVA02-S In22k In1k | eva02_small_patch14_336.mim_in22k_ft_in1k | timm / huggingface |
| DeiT3 | deit3_base_patch16_224 | timm / huggingface |
| DeiT3 In21k In1k | deit3_base_patch16_224_in21ft1k | timm / huggingface |
| DeiT3 FB In22k In1k | deit3_base_patch16_384.fb_in22k_ft_in1k | timm / huggingface |
| DeiT3-L In22k In1k | deit3_large_patch16_384.fb_in22k_ft_in1k | timm / huggingface |

## M. Use of AI Assistance

AI assistants, such as ChatGPT, were utilized in various aspects of the research, including coding, data analysis, and writing tasks. These tools helped automate repetitive tasks, generate initial drafts, and assist in exploring potential solutions. However, all AI-generated outputs were reviewed and refined by researchers to ensure accuracy and coherence.

