# OpenReview forum: "A Geometry-Based View of Mahalanobis OOD Detection"
_ICML.cc/2026/Conference — ICML 2026 regular_

### Official Review · Reviewer_bJCr · 2026-03-01

**Soundness:** 3
**Presentation:** 3
**Significance:** 3
**Originality:** 3
**Overall Recommendation:** 4
**Confidence:** 4

**Summary:**

This paper examines the strong representation dependence of Mahalanobis-based out-of-distribution (OOD) detection in modern vision models from a geometric perspective. The authors show through systematic cross-model experiments that OOD performance varies substantially across architectures and training regimes and is not reliably predicted by classification accuracy. The submission investigates the concept that in-distribution (ID) feature geometry governs detector behavior, identifying local intrinsic dimensionality and within-class spectral slope as complementary signals whose product provides a compact ID-only predictor of performance. Based on this insight, the paper proposes a radially scaled ℓ2 normalization parameterized by β to reshape feature radii while preserving directions, together with an ID-only β-selection strategy. Experiments demonstrate consistent improvements over fixed normalization baselines and performance approaching oracle tuning. Overall, the authors investigate an important context for understanding how feature geometry and normalization influence the reliability of Mahalanobis-style OOD detection.

**Compliance With Llm Reviewing Policy:**

Affirmed.

**Key Questions For Authors:**

1.The study focuses on vision Transformers; have the authors evaluated whether the geometric proxy and radial scaling generalize to other architectures (e.g., CNNs)? It may also be helpful to briefly relate the approach to recent test-time adaptation methods for domain generalization (e.g., Test Time Prompt Adaptation for Domain Generalization, IJCNN 2025).\
2.The m·|s| proxy appears primarily supported by empirical correlation; were stronger theoretical justifications or alternative geometric candidates considered? A deeper theoretical grounding would reinforce the conceptual contribution.\
3.The ID-only β selection relies on LID and spectral estimation; could the authors clarify its computational complexity and scalability, and further interpret the model-dependent β regimes geometrically? This would improve both practical and conceptual clarity.

**Limitations:**

yes

**Strengths And Weaknesses:**

Strengths:\
1.The paper combines systematic empirical evaluation with a coherent geometric and spectral analysis that mechanistically explains Mahalanobis behavior.\
2.The technical exposition is clear and well structured, with precise metric definitions and reproducible experimental setup.\
3.The work establishes a quantitative link between OOD performance and measurable in-distribution geometry, clarifying representation–detector alignment.\
4.The radial scaling formulation with ID-only β selection provides a conceptually unified and technically meaningful extension of prior Mahalanobis-based approaches.\
Weaknesses:\
1.The empirical study is largely limited to vision Transformers and Mahalanobis-type detectors, leaving the generality of the geometric proxy and radial scaling beyond these settings unclear.\
2.The stability analysis is technically dense and could benefit from clearer intermediate explanations and deeper discussion of model-dependent β regimes.\
3.While the m·|s| proxy shows strong empirical correlation, a more rigorous theoretical justification of its optimality and broader applicability remains open.\
4.The radial scaling formulation can be viewed as a parametric extension of ℓ2 normalization, making the methodological novelty incremental rather than fundamental.

---

> ### Author Rebuttal · Authors · 2026-03-31
>
> > **W1. The paper is validated mostly on ViTs and Mahalanobis-style detectors, so generality beyond these settings is unclear.**
>
> The empirical study focuses on transformer backbones to keep the representation family controlled while analyzing how pretraining and fine-tuning affect geometry (see Section 4,5 and Appendix D). To test transfer beyond Mahalanobis-style quadratic detectors, we evaluated KNN with Mahalanobis-chosen $\beta$ values (**oracle regret** $|\mathrm{FPR}(\hat{\beta})-\mathrm{FPR}(\beta^*)|$, which measures the FPR cost (in percentage points) of using the selected $\hat{\beta}$ instead of the oracle-best $\beta$, and mean FPR@95; lower is better, setup as in Table 1).
>
> | Selector | MeanReg | P90Reg | MaxReg | Mean FPR@95 |
> | --- | --- | --- | --- | --- |
> | MD oracle $\rightarrow$ KNN | **1.03** | **3.08** | **11.68** | **38.12** |
> | MD proxy $\rightarrow$ KNN | 2.31 | 6.07 | 11.65 | 39.40 |
> | $\beta=1$ | 3.06 | 8.04 | 19.76 | 40.15 |
> | $\beta=0$ | 3.57 | 8.47 | 16.83 | 40.66 |
>
> MD-based transfer is promising: the MD proxy improves over fixed $\beta=1$ and $\beta=0$
>
> > **W2. The stability analysis is dense and needs clearer intermediate explanations and discussion of model-dependent beta regimes.**
> We agree and will revise this section to proceed from intuition to derivation: first the exact score factorization into size and stretch, then the instability decomposition and compensation term, and only then the full derivation.
>
> Geometrically, different models occupy different $\beta$ regimes because radial scaling changes residual size and whitening-induced anisotropy simultaneously. Some representations exhibit an interior tradeoff, while others are effectively endpoint-like. This explains why the proxy curve can be inverted-U for some models and U-shaped for others, for example ViT in Figure 9. We will make this model dependence explicit and clarify that the turning-point rule is intended to extract the dominant interior transition rather than assume one universal curve shape.
>
> > **W3. The $m \cdot |s|$ proxy is empirically strong, but its optimality and broader applicability remain theoretically under-justified.**
>
> We do not claim that $m \cdot |s|$ is provably optimal. It is a coefficient-free empirical proxy motivated by the theoretical decomposition. Eq. (15) identifies the relevant geometric channels, size and stretch, but the full expression also contains a compensation term, which can introduce noise for selection.
>
> Below is oracle-regret analysis under a supplementary model-averaged reduced-grid protocol:
>
> | Selector | MeanReg | P90Reg | MaxReg |
> | --- | --- | --- | --- |
> | **Proxy: $m_{k}\lvert s \rvert$** | **1.47** | **4.32** | **8.32** |
> | Eq. (15) without compensation | 2.23 | 6.59 | 13.83 |
> | Full Eq. (15): $I_{\text{total}}$ | 3.75 | 9.88 | 15.67 |
>
> The proxy outperforms both Eq. (15)-based alternatives on every regret metric. This supports the intended interpretation: Eq. (15) is most useful as an explanatory decomposition, while the proxy is the practical selector for robust ID-only tuning. Among the alternatives we evaluated in the paper, the proxy achieves the lowest oracle regret (Figure 4).
>
> > **W4. Radial scaling may look like only an incremental extension of $\ell_2$ normalization, so the novelty can appear limited.**
>
> We agree that radial scaling alone should not be presented as the main novelty. The contribution is the integrated pipeline: identifying a geometry-dependent explanation of Mahalanobis behavior, linking it to size/stretch instability, and using radial scaling with an ID-only selector to approach oracle tuning without OOD access.
>
> > **Q1. Do the geometric proxy and radial scaling generalize beyond ViTs, and how do they relate to test-time adaptation / domain generalization methods?**
>
> Architecture and detector generality are addressed in W1 above. Regarding test-time adaptation and domain generalization, these methods address a different setting: they adapt prompts or model behavior at test time, whereas our method is a post-hoc representation-space transformation with an ID-only selector for $\beta$.
>
> > **Q2. Did the authors consider stronger theoretical justifications or alternative geometric candidates for the $m \cdot |s|$ proxy?**
>
> Addressed in W3 above
>
> > **Q3. What is the computational complexity and scalability of ID-only beta selection, and how should one interpret model-dependent beta regimes geometrically?**
>
> The computational cost is incurred once offline on ID validation features not at OOD evaluation time. For each candidate $\beta$, we compute transformed features, estimate LID from neighborhood structure, and estimate the spectral statistic from the covariance eigenspectrum. The cost therefore scales with the number of ID validation samples, feature dimension, and beta-grid size, making it a modest offline model-selection step rather than a per-sample overhead. See W2 for the geometric interpretation of model-dependent $\beta$ regimes.

---

> > ### Author Rebuttal · Reviewer_bJCr · 2026-04-05
> >
> > I have no concerns. I have decided to keep my score.

---

### Official Review · Reviewer_xnZa · 2026-03-12

**Soundness:** 3
**Presentation:** 3
**Significance:** 3
**Originality:** 3
**Overall Recommendation:** 4
**Confidence:** 4

**Summary:**

This paper investigates out-of-distribution detection methods based on the Mahalanobis distance. It finds that the performance of these approaches is highly dependent on the in-distribution geometric characteristics of the feature space. Moreover, their effectiveness varies significantly across different pre-trained representations, pre-training datasets, and fine-tuning strategies. This paper associates performance variations with the geometric characteristics of the ID feature space. It identifies two principal indicators, the within-class spectral structure and the local intrinsic dimensionality, both of which demonstrate strong effectiveness in characterizing the performance of Mahalanobis distance based OOD detection methods.

**Compliance With Llm Reviewing Policy:**

Affirmed.

**Final Justification:**

The response has addressed my main concerns. My final score is 4 (weak accept).

**Key Questions For Authors:**

1. Near-OOD and far-OOD scenarios may impose different requirements on feature space geometry. Since the current $\beta$ selection criterion is validated only in far-OOD settings, it remains unclear whether the strategy remains stable and effective in more challenging near-OOD scenarios. Could the authors provide additional experiments to verify the robustness of the $\beta$ selection rule under near-OOD conditions?
2. Could the authors provide a formal theoretical justification for the $\beta$ selection rule that chooses the interior point farthest from the endpoint baselines, clarifying why this point corresponds to the optimal OOD detection performance?
3. How do the proposed RS-MD and RS-RMD compare with recent state-of-the-art methods in the OOD detection literature in terms of performance? More broadly, do these approaches demonstrate sufficient competitiveness within the broader landscape of OOD detection 4. The paper adopts K=50 as the default setting, yet it remains unclear whether variations in K would influence the performance of the proposed method. It is recommended to include additional ablation studies to evaluate the sensitivity of the method to different choices of K.
5. The abbreviation LID appears multiple times prior to its formal definition in Section 6.4. It is recommended to introduce the term local intrinsic dimensionality (LID) at its first occurrence to improve clarity and readability for the audience.

**Limitations:**

Please see Weaknesses.

**Strengths And Weaknesses:**

Strengths：
1. The paper does not confine itself to methodological tuning when addressing the performance instability of Mahalanobis distance based detectors, but instead investigates the fundamental causes of performance fluctuations from the perspective of feature geometry.
2. The experimental evaluation spans diverse model architectures, pre-training paradigms, and OOD benchmarks, strictly adhering to the OpenOOD evaluation protocol, thereby ensuring strong data integrity and reproducibility.
3. The paper does not introduce its methodological innovations and theoretical analyses as independent or disconnected contributions. It first uncovers the relationship between geometric properties and detection performance through empirical analysis, and then develops a radially scaled normalization approach based on these findings. Finally, the paper employs the Unified Stability Theory framework to provide a systematic theoretical interpretation of the underlying mechanism, thereby establishing a complete paradigm that integrates empirical discovery, methodological design, and theoretical validation.

Weeknesses:
1. The study does not distinguish between near-OOD scenarios, where distributions closely resemble ID data, and far-OOD scenarios. As a result, it is unable to validate the stability of the $\beta$ selection criterion under more challenging near-OOD scenarios, which constitute the primary practical bottleneck in real-world deployment.
2. The proposed interior turning point selection rule, which chooses the interior point farthest from the endpoint baselines, is a heuristic strategy derived from empirical observations. It lacks a formal theoretical justification demonstrating that the selected point corresponds to the optimal OOD detection performance.
3. The evaluation compares the proposed method only with classical baselines, as well as variants of Mahalanobis distance based approaches. It does not include performance comparisons with recent state-of-the-art OOD detection methods, making it difficult to assess the method’s competitiveness within the broader OOD detection literature and limiting the conclusions to advantages over Mahalanobis distance based techniques.
4. The number of neighbors for LID estimation is fixed at K=50 without conducting ablation studies on different values, making it difficult to assess the sensitivity of the proposed method to this hyperparameter.

---

> ### Author Rebuttal · Authors · 2026-03-31
>
> > **W1. The study does not distinguish between near-OOD and far-OOD, so it is unclear whether the selection rule remains stable in the more difficult near-OOD regime.**
>
> We added a dedicated near (NINCO, SSB-Hard) vs. far (OpenO, Textures, iNaturalist) evaluation. Since our contribution is an ID-only selector for $\beta$ rather than a new detector architecture, raw FPR@95 alone cannot fully answer this question because it conflates detector quality with selector quality. We therefore also report oracle regret: $|\mathrm{FPR}(\hat{\beta}) - \mathrm{FPR}(\beta^*)|$ which measures the FPR cost of using the selected $\hat{\beta}$ instead of the oracle-best  $\beta$. We report MeanReg, P90Reg, and MaxReg.
>
> The oracle-regret analysis shows that the proxy’s advantage is **larger in the near-OOD regime**, precisely where tuning matters most:
>
> | OOD Split | Selector | MeanReg | P90Reg | MaxReg |
> | --- | --- | --- | --- | --- |
> | Near | Proxy | 2.54 | 6.73 | 12.01 |
> | Near | $\beta = 1$ | 3.49 | 9.43 | 16.24 |
> | Far | Proxy | 1.63 | 4.15 | 14.72 |
> | Far | $\beta = 1$ | 2.28 | 6.12 | 25.39 |
>
> > **W2. The interior turning-point rule is heuristic and lacks formal justification.**
>
> We do not claim theoretical optimality for the turning-point rule. Its role is narrower: a practical, coefficient-free way to extract $\hat{\beta}$ from the ID-only proxy curve $P(\beta) = m_k(\beta)|s(\beta)|$.
> The rule is needed because the proxy curve is not uniformly monotone across models. It is inverted-U for some architectures and U-shaped for others (e.g., ViT; see Figures 7 and 9). This model dependence follows from the size-vs-stretch factorization of the Mahalanobis score: radial scaling simultaneously changes residual magnitude and whitening-induced anisotropy, so a fixed “always maximize” or “always minimize” rule is inappropriate.In our ablation against naive max/min extraction, the turning-point rule yields the lowest oracle regret, supporting it as a practical heuristic; we will include this comparison explicitly in the revision heuristic.
>
> > **W3. The evaluation does not compare against recent state-of-the-art OOD methods, so competitiveness is unclear.**
>
> We expanded the evaluation to include recent OOD detectors under the same backbone and protocol: **SCALE** (Xu et al., ICLR 2024), **FDBD** (Liu & Qin, ICML 2024), and **NCI** (Liu & Qin, CVPR 2025), in addition to MSP, MLS, KNN, and ViM.
>
> | Method | Mean FPR@95 |
> | --- | --- |
> | **RS-MD (ours)** | **35.3** |
> | RS-RMD (ours) | 35.8 |
> | MD++ | 36.0 |
> | RMD++ | 36.1 |
> | FDBD | 37.7 |
> | MD | 37.2 |
> | RMD | 37.2 |
> | NCI | 42.2 |
> | SCALE | 49.2 |
>
> RS-MD achieves the best mean FPR@95 among all compared methods, and RS-RMD is second-best.
>
> > **W4. The choice of $K=50$ for LID is not ablated, so sensitivity is unclear.**
>
> We added a k-ablation over $k \in \{10,25,50,100\}$, again evaluated with oracle regret.
>
> | Detector | k | MeanReg | P90Reg |
> | --- | --- | --- | --- |
> | MD | 10 | 2.28 | 5.50 |
> | MD | 25 | 2.04 | 5.38 |
> | MD | 50 | 1.99 | 5.44 |
> | MD | 100 | 2.75 | 5.75 |
> | RMD | 10 | 1.17 | 2.46 |
> | RMD | 25 | 1.26 | 2.63 |
> | RMD | 50 | 0.93 | 2.30 |
> | RMD | 100 | 0.82 | 2.14 |
>
> For MD, MeanReg varies only modestly and P90Reg stays tightly clustered. For RMD, performance improves steadily as k increases. We therefore use k=50 as a practical default because it provides strong overall selector quality, not because the method is uniquely optimized at that value. The preferred k can depend on the evaluation criterion, but the method is robust across the tested range.
>
> > **Q1. Could the authors provide additional experiments to verify robustness under near-OOD conditions?**
>
> Addressed in W1 above. We now report separate near-OOD and far-OOD results with oracle-regret metrics, showing that the selector remains effective and in fact shows its largest regret reductions in the harder near-OOD regime.
>
> > **Q2. Could the authors provide a formal theoretical justification for the interior turning-point rule?**
>
> Addressed in W2 above. We do not claim a formal optimality theorem; instead, we provide a geometric motivation rooted in the size-vs-stretch factorization. Empirically, the turning-point rule yields substantially lower regret than naive max/min alternatives.
>
> > **Q3. How do RS-MD and RS-RMD compare with recent state-of-the-art OOD methods?**
>
> Addressed in W3 above. On the expanded benchmark, RS-MD achieves the best mean FPR@95 and RS-RMD is second-best among all compared methods, including recent detectors from ICLR 2024, ICML 2024, and CVPR 2025.
>
> > **Q4. Would varying K influence the performance of the proposed method?**
>
> Addressed in W4 above. The k-ablation shows the method is robust across k in {10, 25, 50, 100}; k=50 is a practical default, not a specially tuned value.
>
> > **Q5. LID appears before it is formally defined.**
>
> Thank you. We will define “local intrinsic dimensionality (LID)” at its first occurrence in the main text.

---

> > ### Author Rebuttal · Reviewer_xnZa · 2026-04-02
> >
> > Thank you very much for the reviewer's response. It has resolved my doubts and I will maintain my original score

---

### Official Review · Reviewer_2xC5 · 2026-03-14

**Soundness:** 3
**Presentation:** 3
**Significance:** 3
**Originality:** 3
**Overall Recommendation:** 4
**Confidence:** 4

**Summary:**

This paper investigates why Mahalanobis-based out-of-distribution (OOD) detectors perform inconsistently across modern pretrained vision representations. While Mahalanobis scoring remains a strong classical baseline for OOD detection, the authors argue that its effectiveness is largely determined by the geometry of the learned feature space rather than the detector itself.

The authors conduct a large-scale empirical study across multiple foundation-model backbones and Mahalanobis-style detectors. They show that OOD detection performance varies substantially depending on the pretrained representation and training regime. To explain this variability, the paper introduces a geometric characterization of the in-distribution (ID) feature space using two quantities: within-class spectral structure and local intrinsic dimensionality (LID). These measures are shown to correlate strongly with OOD detection performance.

Motivated by this geometric perspective, the authors propose radially scaled normalization, which preserves feature directions while modifying feature radii. Adjusting the scaling parameter \beta  effectively alters the geometry of the representation space without retraining the backbone. The authors demonstrate that selecting \beta based on ID-only geometric signals can improve OOD detection performance compared with standard normalization baselines.

Overall, the paper argues that Mahalanobis OOD detection performance  depends on representation geometry, and experimental results demonstrate that the proposed simple normalization-based transformations typically outperform baselines without normalization.

**Compliance With Llm Reviewing Policy:**

Affirmed.

**Key Questions For Authors:**

1) In Figure 7,  for the ViT model, the largest slopxLID appears when \beta is around 2. Why \beta is selected to be the star around 0?  Is there a typo or I have a misunderstanding here?

2). Eq. (15) provides an indirect  theoretical justification for the prediction of \beta using the simple product m(\beta)\vert s(\beta)\vert. Is it possible to use Eq. (15) directly to predict \beta and compare their performances?

**Limitations:**

This paper is mainly an empirical analysis paper with simple adjustment (i.e., normalization with selected scales) to improve performance, but no major algorithmic novel contribution.

**Strengths And Weaknesses:**

This paper provides an interesting geometric perspective on Mahalanobis-based OOD detection and identifies representation properties that influence detector performance. The proposed normalization technique is simple and typically improves performance in comparison to using original extracted features. Theoretical analysis in Eq (15)  provides a principled rationale for the effectiveness of predicting the scale \beta using the simple product m(\beta) \vert s(\beta)\vert.   However,  this theoretical justification is indirect and weak. It is not clear whether using Eq (15) directly to select the scale \beta can improve the performance as well.

---

> ### Author Rebuttal · Authors · 2026-03-31
>
> We thank the reviewer for the thoughtful and constructive feedback. Below is our response.
>
> > **W1. Eq. (15) gives only an indirect and weak theoretical rationale for using $m(\beta)|s(\beta)|$, and it is unclear whether using Eq. (15) directly would work better.**
> >
>
> Thank you for this point. We want to be precise about the relationship between Eq. (15) and the proxy. Our claim is not that Eq. (15) directly derives the selector. Rather, Eq. (15) is valuable because it identifies the main geometric channels driving Mahalanobis instability: a *size channel* (how residual magnitude changes with $\beta$) and a *stretch channel* (how whitening-induced anisotropy redistributes the residual). The full instability decomposition also contains a *compensation term* that can partially cancel these contributions, which is precisely why optimizing Eq. (15) directly is less effective than tracking the underlying drivers before cancellation.
>
> The proxy $m_k(\beta)|s(\beta)|$ is intended to capture these two drivers in a coefficient-free form: $|s(\beta)|$ tracks within-model spectral heterogeneity, while $m_k(\beta)$ helps normalize across models. The multiplicative form is also geometrically appropriate: when $|s(\beta)| \approx 0$, whitening is close to isotropic, and a product correctly suppresses the proxy, whereas an additive form would not.
>
> To address the suggestion directly, we compared three selectors on a reduced $\beta$ grid. Because all three are evaluated on the same grid, this isolates selector quality from grid resolution. We evaluate using **oracle regret** $|\mathrm{FPR}(\hat{\beta})-\mathrm{FPR}(\beta^*)|$, which measures the FPR cost (in percentage points) of using the selected $\hat{\beta}$ instead of the oracle-best $\beta$\*. This metric isolates selector quality from detector quality, which is the relevant criterion when the contribution is an ID-only selector rather than a new detector architecture. We report MeanReg (average regret; typical quality), P90Reg (90th-percentile; tail robustness), MaxReg (worst case; catastrophic failure). This is a supplementary model-averaged sanity check, so absolute regret values are not directly comparable to the full benchmark tables, but the selector ranking is.
>
> **Supplementary Eq. (15) selector comparison (model-averaged $\beta$ curves; reduced $\beta$ grid).**
>
> | Selector | MeanReg | P90Reg | MaxReg |
> | --- | --- | --- | --- |
> | **Proxy: $m_{k}\lvert s \rvert$** | **1.47** | **4.32** | **8.32** |
> | Eq. (15) without compensation | 2.23 | 6.59 | 13.83 |
> | Full Eq. (15): $I_{\text{total}}$ | 3.75 | 9.88 | 15.67 |
>
> The proxy outperforms both Eq. (15)-based alternatives on every regret metric. Removing the compensation term improves selection quality, which is consistent with the view that this term makes direct optimization less stable, but the simpler proxy remains superior. This supports our intended interpretation: Eq. (15) is most useful as an explanatory decomposition identifying the relevant geometric channels, while the proxy is the practical selector designed for robust ID-only tuning.
>
> ---
>
> > **Q1. In Figure 7, for ViT, the largest slope×LID appears around $\beta \approx 2$. Why is the selected $\beta$ starred around 0?**
> >
>
> This is not a typo. The star denotes the selected $\hat{\beta}$ from our **interior turning-point rule**, not the global extremum of the raw curve. The rule selects the most pronounced interior turning point relative to the endpoint reference level, which returns an interior maximum for inverted-U curves and an interior minimum for U-shaped curves.
>
> ViT belongs to the latter case. As our instability analysis shows (Figure 9), the proxy trajectory is model-dependent: ViT exhibits a U-shaped profile with an interior minimum near $\beta \approx 0$, while other models (BEiTV2, CLIP) follow the opposite trend. This model dependence is expected from the size-vs-stretch analysis: different representations balance residual magnitude and whitening-induced anisotropy differently, producing qualitatively different proxy curves. The turning-point rule selects this interior minimum, deliberately avoiding the boundary value at $\beta \approx 2$, which may be an artifact of the finite search range rather than a genuinely optimal operating point.
>
> We will clarify in both the main text and figure caption that the star marks the turning-point selection, not the boundary extremum, and explicitly note that ViT falls into the U-shaped regime.
>
> ---
>
> > **Q2. Is it possible to use Eq. (15) directly to predict $\beta$ and compare performance?**
> >
>
> Yes, we tested this directly, and the proxy outperforms both the full Eq. (15) and Eq. (15) without compensation on all regret metrics and in proxy-FPR correlation. Please see our response to W1 above for the full comparison table and discussion.

---

> > ### Author Rebuttal · Reviewer_2xC5 · 2026-04-07
> >
> > Thanks to the authors for their detailed explanations. I maintain my positive score.

---

### Official Review · Reviewer_YcTm · 2026-03-18

**Soundness:** 3
**Presentation:** 3
**Significance:** 3
**Originality:** 3
**Overall Recommendation:** 4
**Confidence:** 1

**Summary:**

This paper conducts an empirical and geometric analysis of Mahalanobis-based OOD detection across diverse pretrained representations. It shows that OOD performance varies significantly with the underlying feature space and links this variability to intrinsic geometric properties, including within-class spectral structure and local intrinsic dimensionality. Based on this observation, the paper proposes a radial normalization scheme that rescales feature norms while preserving directions, and selects its scaling parameter using ID-only geometric signals, aiming to improve OOD detection without modifying the backbone.

**Compliance With Llm Reviewing Policy:**

Affirmed.

**Key Questions For Authors:**

Please refer to the weaknesses above. Moreover, how does the proposed method generalize to other OOD detectors beyond Mahalanobis-style methods?

**Limitations:**

Yes.

**Strengths And Weaknesses:**

**Strengths**

-  Understanding when Mahalanobis OOD detection works is valuable for deploying pretrained vision models reliably.
- Connecting OOD performance to feature geometry  provides a potentially useful diagnostic viewpoint.
- The proposed radial normalization does not require retraining the backbone.

**Weaknesses**

- The central claim that the proposed geometric proxy (e.g., LID × spectral slope) reliably predicts OOD performance is not fully convincing based on the provided evidence (e.g., Fig. 5 lacks clear quantitative validation or statistical analysis).
- The method for estimating or selecting geometric factors is not thoroughly validated; more controlled experiments are needed.
- While the paper observes that different representations yield different OOD performance, it does not provide sufficient theoretical or empirical analysis explaining why these differences arise.
- The proxy-selected normalization parameter shows limited gains compared to simple fixed normalization (e.g., β=1), raising questions about the practical benefit of the proposed selection strategy.

---

> ### Author Rebuttal · Authors · 2026-03-30
>
> We thank the reviewer for the thoughtful and constructive feedback. Below is our response.
>
> > **W1. The central claim that the proposed geometric proxy reliably predicts OOD performance is not fully convincing.**
>
> To complement Fig. 5, we now report quantitative correlations between the proxy and OOD FPR across models:
>
> | Detector | $k$ | Pearson $r$ | Spearman $\rho$ |
> | --- | --- | --- | --- |
> | MD | 10 | -0.58 | -0.74 |
> | MD | 25 | -0.83 | -0.79 |
> | MD | 50 | -0.84 | -0.85 |
> | MD | 100 | -0.82 | -0.77 |
> | RMD | 10 | -0.4 | -0.51 |
> | RMD | 25 | -0.67 | -0.64 |
> | RMD | 50 | -0.77 | -0.72 |
> | RMD | 100 | -0.34 | -0.44 |
>
> Correlations are consistently negative, especially for MD and intermediate k, supporting that larger proxy values are associated with lower FPR. All MD entries and most intermediate-k RMD entries are statistically significant.
>
> > **W2. The method for estimating or selecting geometric factors is not thoroughly validated.**
>
> Since our contribution is an ID-only selector for $\beta$, we evaluate selector quality using oracle regret, $|\mathrm{FPR}(\hat{\beta})-\mathrm{FPR}(\beta^*)|$, rather than raw FPR alone.  We report MeanReg (average regret; typical quality) and P90Reg (90th-percentile; tail robustness).
>
> First, the proxy $m \times |s|$ is not brittle to the LID neighborhood size:
>
> | Detector | $k$ | MeanReg | P90Reg |
> | --- | --- | --- | --- |
> | MD | 10 | 2.28 | 5.50 |
> | MD | 25 | 2.04 | 5.38 |
> | MD | 50 | 1.99 | 5.44 |
> | MD | 100 | 2.75 | 5.75 |
> | RMD | 10 | 1.17 | 2.46 |
> | RMD | 25 | 1.26 | 2.63 |
> | RMD | 50 | 0.93 | 2.30 |
> | RMD | 100 | 0.82 | 2.14 |
>
> Second, the multiplicative proxy outperforms the individual factors and the additive form:
>
> | Proxy | MeanReg | P90Reg |
> | --- | --- | --- |
> | $m \times \lvert s \rvert$ | **1.99** | **5.44** |
> | $m$ | 2.65 | 7.01 |
> | $m + \lvert s \rvert$ | 2.65 | 7.01 |
> | $\lvert s \rvert$ | 5.42 | 18.48 |
>
> Importantly, $m + |s|$ does not improve over $m$ alone, whereas the multiplicative form does.
>
> Third, among the alternatives we evaluated in the paper, the proxy achieves the lowest oracle regret (Figure 4).
>
>
> > **W3. The paper does not sufficiently explain why different representations yield different OOD performance.**
>
> The Mahalanobis score factorizes into a size term, residual magnitude, and a stretch term, whitening redistribution across eigendirections. Different representations therefore differ in both residual-size variability and spectral heterogeneity, and $\beta$ changes the balance between these two channels. This also explains the per-direction results: after inverse-variance weighting, low-variance directions can dominate, so directions important for OOD detection need not be those with the largest raw variance. The full instability decomposition contains a compensation term that partially cancels size and stretch. The simpler proxy tracks the two main drivers before cancellation, which is why it outperforms the full instability expression.  In short, different representations yield different OOD behavior because radial scaling changes both residual norm dispersion and spectral anisotropy, and the relative dominance of these two effects differs across models.
>
> > **W4. The proxy-selected normalization parameter shows limited gains compared to fixed $\beta=1$.**
>
> We agree the practical claim should be narrower. The benefit is not universal large gains over a fixed baseline, but reliable ID-only tuning when the optimal normalization varies across datasets.
>
> | Dataset | Selector | MeanReg | P90Reg | MaxReg |
> | --- | --- | --- | --- | --- |
> | NINCO | **Proxy** | **1.92** | **3.99** | **8.32** |
> | NINCO | $\beta=1$ | 2.56 | 6.32 | 8.75 |
> | SSB-Hard | **Proxy** | **3.16** | **7.34** | **12.01** |
> | SSB-Hard | $\beta=1$ | 4.41 | 12.10 | 16.24 |
> | OpenImage-O | **Proxy** | **0.73** | **1.68** | **2.77** |
> | OpenImage-O | $\beta=1$ | 1.10 | 2.38 | 3.83 |
> | iNaturalist | **Proxy** | **2.24** | **6.21** | **14.72** |
> | iNaturalist | $\beta=1$ | 3.34 | 8.42 | 25.39 |
>
> Proxy selection reduces regret relative to fixed $\beta=1$ on every dataset. The gains are especially clear on harder cases such as SSB-Hard and iNaturalist, where the proxy avoids catastrophic mistuning.
>
> > **Q1. How does the proposed method generalize to detectors beyond Mahalanobis-style methods?**
>
> We tested this directly by measuring KNN transferability of Mahalanobis-chosen $\beta$ (oracle regret and mean FPR@95; lower is better, setup as in Table 1).
>
> | Selector | MeanReg | P90Reg | MaxReg | Mean FPR@95 |
> | --- | --- | --- | --- | --- |
> | MD oracle $\rightarrow$ KNN | **1.03** | **3.08** | **11.68** | **38.12** |
> | MD proxy $\rightarrow$ KNN | 2.31 | 6.07 | 11.65 | 39.40 |
> | $\beta=1$ | 3.06 | 8.04 | 19.76 | 40.15 |
> | $\beta=0$ | 3.57 | 8.47 | 16.83 | 40.66 |
>
> MD-based transfer works well: the MD proxy improves over fixed $\beta=1$ and $\beta=0$. We view this as preliminary detector-transfer evidence for one non-Mahalanobis detector, not a detector-agnostic guarantee.

---

> > ### Author Rebuttal · Reviewer_YcTm · 2026-04-07
> >
> > Thanks for the authors' detailed response. My main concerns have been well resolved.

---

### Decision · Program_Chairs · 2026-04-30

**Decision:**

Accept (regular)

**Comment:**

The reviewers are in general agreement regarding the paper's strengths, highly commending the originality of the analysis that elucidates performance variability in Mahalanobis-based OOD detection through the lens of feature geometry, such as local intrinsic dimensionality and spectral properties. The proposed practical improvement scheme, which requires no backbone retraining and is validated by extensive experiments, was also well-received. On the other hand, concerns persist regarding the lack of rigorous theoretical justification and proofs of optimality for the proposed metrics. Furthermore, reviewers pointed out the insufficient comparison with recent state-of-the-art methods and questioned the generalizability to near-OOD scenarios and other architectures. While the rebuttal addressed several concerns, it did not fundamentally shift the reviewers' overall assessments. Nevertheless, given the clear contributions of the work, the AC is pleased to recommend acceptance.

Additionally, please address the following discrepancy:
- The title of the paper in the PDF does not match the title registered on OpenReview. Please verify and ensure that the titles are consistent.
- Regarding the reference below, please note that the authors and part of the title differ from those in the linked arXiv version. Please verify and correct the entry.
Zhao, S., Liu, J., Wen, X., Tan, H., and Qi, X. Rethinking out-of-distribution detection in vision foundation models, October 2024b. URL https://arxiv.org/abs/2302.02615.